# Intracellular functions and motile properties of bi-directional kinesin-5 Cin8 are regulated by neck linker docking

Alina Goldstein-Levitin[1], Himanshu Pandey[1], Kanary Allhuzaeel[1], Itamar Kass[1,2], Larisa Gheber[1]*

[1]Department of Chemistry, Ben-Gurion University of the Negev, Beer-Sheva, Israel; [2]InterX LTD, Ramat-Gan, Israel

**Abstract** In this study, we analyzed intracellular functions and motile properties of neck-linker (NL) variants of the bi-directional *S. cerevisiae* kinesin-5 motor, Cin8. We also examined – by modeling – the configuration of H-bonds during NL docking. Decreasing the number of stabilizing H-bonds resulted in partially functional variants, as long as a conserved backbone H-bond at the N-latch position (proposed to stabilize the docked conformation of the NL) remained intact. Elimination of this conserved H-bond resulted in production of a non-functional Cin8 variant. Surprisingly, additional H-bond stabilization of the N-latch position, generated by replacement of the NL of Cin8 by sequences of the plus-end directed kinesin-5 Eg5, also produced a nonfunctional variant. In that variant, a single replacement of N-latch asparagine with glycine, as present in Cin8, eliminated the additional H-bond stabilization and rescued the functional defects. We conclude that exact N-latch stabilization during NL docking is critical for the function of bi-directional kinesin-5 Cin8.

*For correspondence:
lgheber@bgu.ac.il

## Introduction

Kinesin-5 motor proteins perform essential mitotic functions by providing the force that separates the spindle poles apart during spindle assembly, maintenance, and elongation reviewed in *Goulet and Moores, 2013*; *Kapoor, 2017*; *Mann and Wadsworth, 2019*; *Scholey et al., 2016*; *Singh et al., 2018*; *Waitzman and Rice, 2014*. These motors are homotetramers, with two pairs of catalytic domains located on opposite sides of a central minifilament (*Acar et al., 2013*; *Gordon and Roof, 1999*; *Kashina et al., 1996*; *Scholey et al., 2014*). This unique structure enables kinesin-5 motors to crosslink the antiparallel microtubules (MTs) of the spindle and to slide them apart by moving in the plus-end direction of the two MTs that they crosslink (*Kapitein et al., 2005*; *Shapira et al., 2017*; *Shimamoto et al., 2015*). Because of the MT architecture within the mitotic spindles, plus-end directed motility of kinesin-5 motors is essential to separate the spindle poles (*Goulet and Moores, 2013*; *Mann and Wadsworth, 2019*; *Singh et al., 2018*). Moreover, since the catalytic domains of kinesin-5 motors are located at their N-termini, it was previously believed that such N-terminal motors are exclusively plus-end directed. Recently, however, several studies have demonstrated independently in single-molecule motility assays that three kinesin-5 motors – the *Saccharomyces cerevisiae* Cin8 and Kip1 and the *Schizosaccharomyces pombe* Cut7 – move processively toward the minus-end of the MTs and switch directionality under different experimental conditions (*Britto et al., 2016*; *Düselder et al., 2015*; *Edamatsu, 2014*; *Fallesen et al., 2017*; *Fridman et al., 2013*; *Gerson-Gurwitz et al., 2011*; *Roostalu et al., 2011*; *Shapira and Gheber, 2016*). The above-described body of work notwithstanding, the molecular mechanism and physiological implications of such bi-directional motility remain elusive.

The majority of kinesin motors share a mechanical structural element of 10–18 amino acids, termed the neck linker (NL), which connects the two catalytic domains of the kinesin dimers, enabling them to step on the same MT. For N-terminal plus-end directed kinesins, the NL is located at the C-terminal end of the catalytic domain, between the last helix of the catalytic domain (α6) and the neck helix (α7) that is required for dimerization (*Kozielski et al., 1997*; *Vale and Fletterick, 1997*). It has been proposed that during 'hand-over-hand' stepping in the plus-end direction, the NL isomerizes between a disordered 'undocked' conformation in the presence of ADP and an ordered, motor-domain-bound 'docked' conformation in the presence of ATP, with the docked conformation pointing toward the plus-end of the MT (*Gigant et al., 2013*; *Goulet et al., 2014*; *Rice et al., 1999*; *Rosenfeld et al., 2001*; *Tomishige et al., 2006*). It is believed that the docked conformation is stabilized by N-terminal sequences upstream of the motor domain, termed the cover strand (CS), which form a β-sheet with the first half of the docked NL, termed β9 (*Geng et al., 2014*; *Goulet et al., 2012*; *Hwang et al., 2008*; *Khalil et al., 2008*; *von Loeffelholz et al., 2019*). Additional stabilization of the docked NL conformation is provided by an asparagine residue in the middle of the NL that is predicted to serve as a latch (N-latch), holding the docked NL along the core motor domain, which is conserved in processive plus-end directed kinesins (*Hwang et al., 2008*). The docked NL is also stabilized by an interaction between amino acids in the second half of the NL, termed β10, and amino acids in β7 and the loop 10 of the motor domain (*Hwang et al., 2008*). The importance of the isomerization of the NL between the docked and undocked conformations – as a force-generating transition in kinesin motors (*Cao et al., 2014*; *Clancy et al., 2011*; *Gigant et al., 2013*; *Goulet et al., 2014*; *Shang et al., 2014*) – is believed to lie in its role in coordinating nucleotide binding and hydrolysis (*Clancy et al., 2011*) and regulating motor velocity, processivity, and force production (*Budaitis et al., 2019*; *Düselder et al., 2012*; *Higuchi and Endow, 2002*; *Hughes et al., 2012*; *Muretta et al., 2015*; *Muretta et al., 2018*; *Schief and Howard, 2001*; *Shastry and Hancock, 2011*).

During plus-end directed stepping of pairs of motor domains connected via the NL, the transition to, and stabilization of, the docked NL conformation bring the trailing head forward in the plus-end direction and are thus critical for plus-end directed motility. Since bi-directional kinesin motors can move in both plus- and minus-end directions on the MTs, the dynamics of the NL isomerization in these kinesins has to allow stepping in both directions. Thus far, however, the role of the NL in regulating the motor functions of bi-directional kinesins has not been reported. To address this issue, we designed a series of NL variants of the bi-directional kinesin-5 motor, Cin8, and examined their functions in vivo and in vitro. We also examined the H-bond arrangement of the docked NL conformations of these NL variants. We showed that, among these variants, those that were partially active exhibited a smaller number of stabilizing H-bonds between the docked NL and motor domain, vs. wild type (wt) Cin8. In addition, we showed that elimination of a conserved backbone H-bond between a glycine in the N-latch position of Cin8 resulted in a non-functional variant, indicating that stabilization of this H-bond is critical for the functionality of Cin8. Partial replacement of the NL of Cin8 with homologous sequences from the NLs of plus-end directed kinesins resulted in the generation of non-functional Cin8 variants that could neither move in the minus-end direction nor crosslink MTs in vitro. In one such variant, containing sequences from the NL of the plus-end directed vertebrate kinesin-5 Eg5, the number of stabilizing H-bonds between the N-latch asparagine and the motor domain was larger than that in wt Cin8, indicating that additional stabilization of the N-latch position is incompatible with the functionality of Cin8. In this variant, a single replacement of the conserved N-latch asparagine with glycine, as originally present in Cin8, decreased the number of N-latch stabilizing H-bonds and rescued the majority of the defects of the non-functional variant, both in vivo and in vitro. Thus, we propose that exact H-bond stabilization that allows a certain degree of flexibility in NL docking is critical for the functionality of the bi-directional kinesin-5 Cin8.

## Results

To examine the role of the NL in regulating the functionality of Cin8, we generated a series of variants in which some of the amino acids of the Cin8 NL sequence were replaced with homologous sequences from other kinesin motors (*Figure 1A and B*). Based on amino acid alignment, we created Cin8 variants containing NL sequences from two exclusively plus-end directed motors, human kinesin 1 KHC (designated here as Cin8$_{NL}$KHC) and *Xenopus laevis* kinesin-5 Eg5 (Cin8$_{NL}$Eg5) or sequences

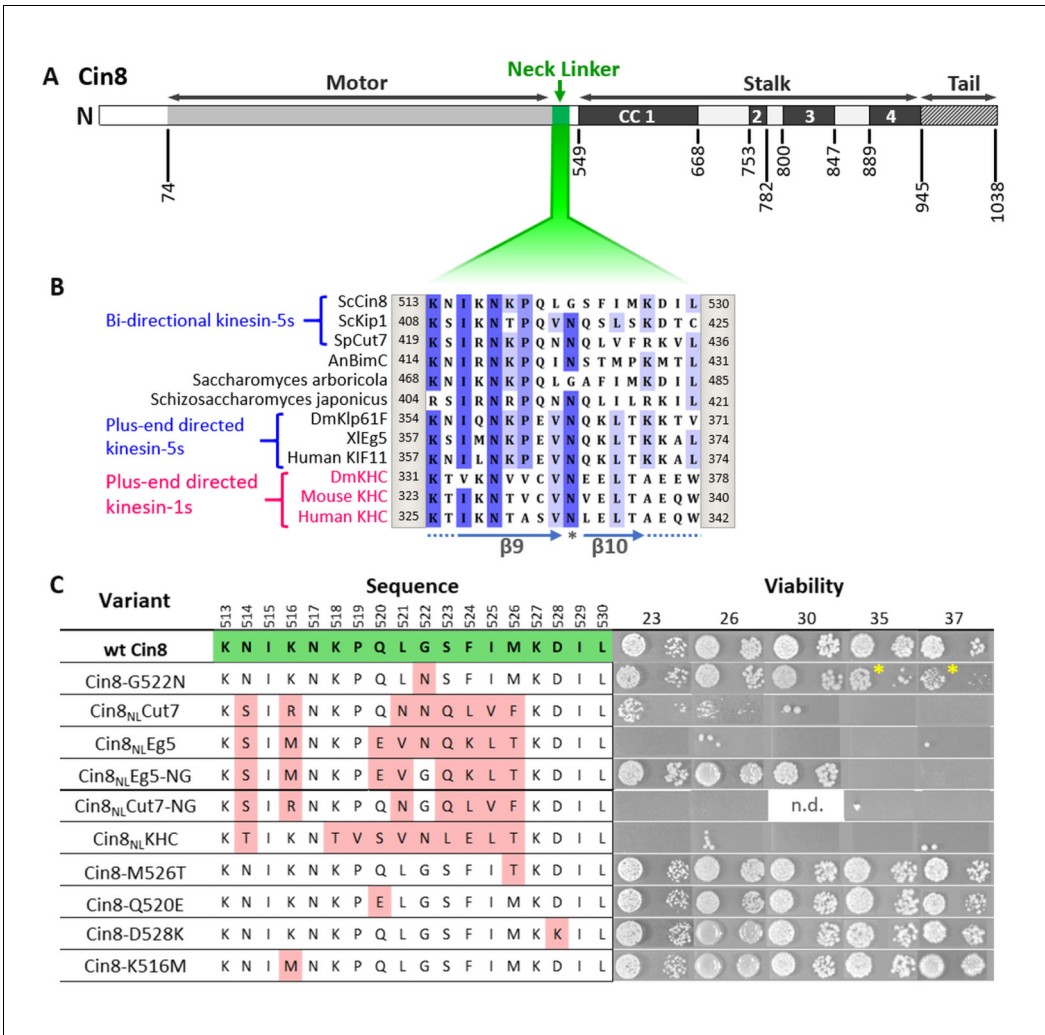

**Figure 1.** Viability of *Saccharomyces cerevisiae* cells expressing NL variants of Cin8. (A) Schematic representation of the Cin8 sequence with amino acid numbers at the bottom flanking the main structural elements of Cin8, indicated on the top; CC: coiled coil. The NL region (green) is expanded in B. (B) Multiple sequence alignment (MSA) of the NL region of members of the kinesin-5 (black) and kinesin-1 (magenta) families. Known directionality of kinesin motors, that is, either bi-directional or exclusively plus end directed, is annotated in blue on the left. The positions flanking the presented sequence of each kinesin motor are annotated on the right and on the left of each sequence. The MSA was calculated by the MUSCLE algorithm (*Edgar, 2004*) via Unipro UGENE program (*UGENE team et al., 2012*). The amino acids are color coded by percentage identity with a 55% threshold. β9 and β10 of the NL are indicated at the bottom of the panel. Asterisk indicates the N-latch position. (C) Viability of *S. cerevisiae* cells expressing NL variants of Cin8, indicated on the left, as a sole source of kinesin-5. Temperatures (°C) at which cell growth was examined are indicated on the top. Amino acids of the NL are indicated in the middle; positions in the sequence of Cin8 are indicated on the top. The amino acids highlighted in green are those of the wt Cin8 sequence. Amino acids in the NL of Cin8 that were mutated to amino acids from other kinesin motors are highlighted in pink. Asterisks indicate the reduced growth of cells expressing the Cin8-G522N variant at 35°C and 37°C. n.d. – not determined.

The online version of this article includes the following figure supplement(s) for figure 1:

**Figure supplement 1.** Multiple sequence alignment (MSA) of the NL region (right) and a phylogenetic tree (left) of kinesin-5 motors.

from the bi-directional *S. pombe* kinesin-5 Cut7 (Cin8~NL~Cut7). In addition, we created Cin8 NL variants in which single amino acids were replaced with amino acids from the above-mentioned kinesin motors (*Figure 1B*). These variants were examined in a series of in vivo and in vitro assays with the aim to study the effect of the above-described replacements on the functionality of Cin8.

# Cin8 variants containing NL sequences from plus-end directed kinesin motors are not functional in cells

To assess the intracellular functionality of the NL variants, we first examined their ability to support yeast viability as the sole source of kinesin-5 function. Since at least one of the two *S. cerevisiae* kinesin-5 motors, Cin8 and Kip1, is essential for cell viability (*Roof et al., 1992*; *Saunders and Hoyt, 1992*), the NL variants were examined on a low-copy centromeric plasmid in cells carrying chromosomal deletions of *CIN8* and *KIP1*, following the shuffling-out of the Cin8 plasmid that covered the double deletion (*Figure 1C*; *Avunie-Masala et al., 2011*; *Goldstein et al., 2019*; *Goldstein et al., 2017*). We found that NL variants carrying a single amino acid replacement at positions where the consensus of most bi-directional kinesin-5 motors differs from the consensus of the exclusively plus-end directed kinesin-5 motors (variants Cin8-K516M, Cin8-Q520E, Cin-M526T, and Cin8-D528K) were mostly viable (*Figure 1C*), thereby indicating these replacements do not significantly affect the functionality of Cin8 in vivo.

In contrast to the single amino acid replacements, Cin8 variants containing partial NL sequences from exclusively plus-end directed kinesin motors (Cin8$_{NL}$KHC and Cin8$_{NL}$Eg5) failed to grow at all the examined temperatures (*Figure 1C*), thereby indicating that these variants were not functional in cells. In addition, cells deleted for Cin8 (but with functional Kip1) and expressing the Cin8$_{NL}$Eg5 variant exhibited significantly longer doubling times than cells expressing wt Cin8 (*Table 1*), as is consistent with the impaired functionality of this variant. Importantly, cells expressing the Cin8 variant with the NL sequence of the bi-directional Cut7 (Cin8$_{NL}$Cut7) were viable at 23°C and 26°C, but not at temperatures above 30°C (*Figure 1C*), thereby indicating that at least some of the Cin8 functions are preserved in the Cin8 variant containing the NL sequence from a different bi-directional kinesin. The doubling time of cells expressing the Cin8$_{NL}$Cut7 variant was significantly longer than that of cells expressing wt Cin8 in both the presence and absence of Kip1 (*Table 1*), which is consistent with the partial functionality of this variant. Moreover, in cells with functional Kip1, variants containing the NL sequence from the plus-end directed Eg5 and bi-directional Cut7 accumulated with monopolar spindles (*Figure 2A, B*), indicating that both variants were defective in the spindle assembly function in *S. cerevisiae* cells.

To elucidate which function of Cin8 is maintained in the partially viable Cin8$_{NL}$Cut7, but not in the non-viable Cin8$_{NL}$Eg5 variant, we visualized the cellular localization of these variants, tagged with 3GFP, in cells containing functional Kip1 and bearing a tdTomato-tagged spindle-pole-body (SPB) protein Spc42 (Spc42-tdTomato) (*Fridman et al., 2013*; *Goldstein et al., 2017*). Prior to bipolar spindle formation, when the two SPBs had not yet separated, wt Cin8 and Cin8$_{NL}$Cut7 concentrated near the SPBs, at the minus-ends of nuclear MTs (*Figure 2C–E*). This localization pattern is consistent with the minus-end directed motility of these variants on the nuclear MTs (*Figure 2D*; *Shapira et al., 2016*). In contrast, Cin8$_{NL}$Eg5 exhibited diffusive localization in the nucleus (*Figure 2C–E*). Quantitative analysis in cells with pre-assembled spindles did indeed indicate that Cin8$_{NL}$Eg5 occupied a significantly larger area than wt Cin8 and Cin8$_{NL}$Cut7 (see Materials and methods) (*Figure 2E*),

**Table 1.** Doubling time of *S. cerevisiae* cells expressing wt and NL variants of Cin8.

| | *cin8Δkip1Δ*[a†] | *cin8Δ*[a] |
|---|---|---|
| wt Cin8 | 152 ± 1 (3) | 127 ± 4 (3) |
| Cin8-G522N | 175 ± 2 (3)* | n.d. |
| Cin8$_{NL}$Cut7 | 210 ± 7 (3)** | 143 ± 3 (4)* |
| Cin8$_{NL}$Eg5-NG | 191 ± 6 (4)** | n.d. |
| Cin8$_{NL}$Eg5 | n.d. | 167 ± 4 (4)** |

[a]Average doubling times ± SEM (min). The number of experiments is shown in parentheses (see also the **Source data 1**). 3HA-tagged Cin8 variants were expressed in the *kip1Δcin8Δ* and *cin8Δ* strains.

[†]In the *cin8Δkip1Δ* strain, NL variants and wt Cin8 were examined following shuffling-out of the parental pMA1208 plasmid (see Materials and methods).

*p < 0.05, **p < 0.01, compared to wt Cin8.

The online version of this article includes the following source data for Table 1:

**Source data 1.** Source data for *Table 1*.

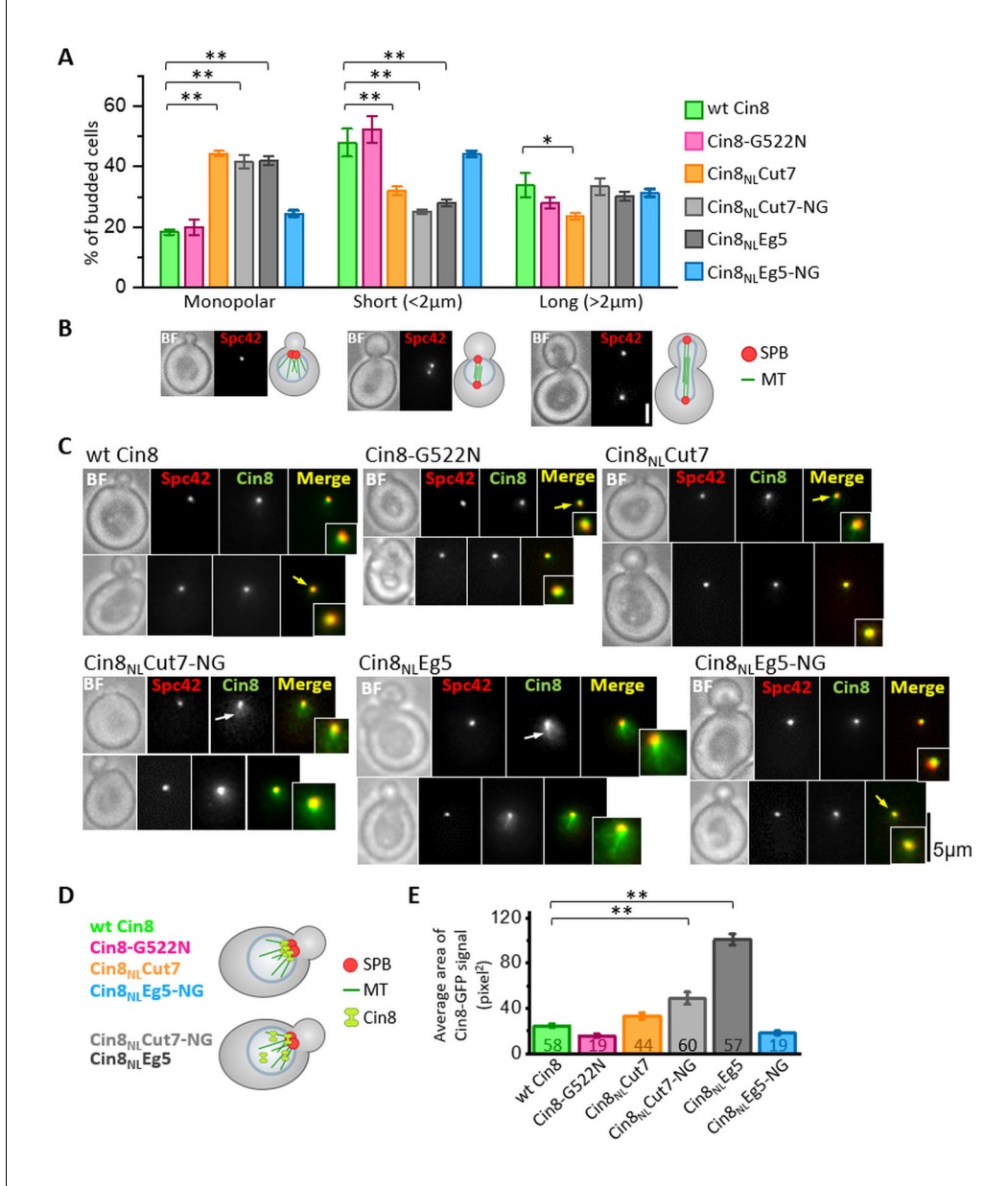

**Figure 2.** Intracellular phenotypes of the NL variants. The examined cells were deleted for the chromosomal copy of *CIN8* (in the presence of *KIP1*) and express tdTomato-tagged SPB component Spc42 and the 3GFP-tagged NL variants. (A and B) Spindle length distribution of cells expressing NL variants of Cin8. (A) The average percentage (± SEM) of budded cells in the different spindle length categories is shown for monopolar, short <2 μm and long >2 μm spindles. Since in *S. cerevisiae* cells the bipolar spindle is formed during the S-phase, budded cells with a short spindle can be either in S-phase or in metaphase. In each experiment, 113–411 cells were examined, and spindles were categorized according to their shapes and lengths (see Materials and methods and *Source data 2*). For each NL variant, three experiments were performed. *p < 0.05; **p < 0.01, compared to wt Cin8. (B) Live cell images (left) and schematic representation of cells and spindles (right) for each spindle category, as in (A). BF: bright field; Bar: 2 μm. (C-E) Localization of NL variants of Cin8 in cells with monopolar spindles. (C) Representative images of cells with small buds and monopolar spindles expressing 3GFP-tagged NL variants of Cin8 (indicated on the top of each panel). Cells were imaged in bright field (BF), red (Spc42) and green (Cin8) fluorescence channels. The insets show a 200% magnification of the localization of Cin8-3GFP. Yellow arrows indicate co-localization of Cin8-3GFP and Spc42-tdTomato, and white arrows indicate localization of Cin8, which is diffusive in the nucleus and is associated with nuclear MTs. Bar: 5 μm. (D) Schematic representation of small budded cells with monopolar spindles showing Cin8-3GFP co-localization with the SPBs (top, as in wt Cin8, Cin8-

*Figure 2 continued on next page*

*Figure 2 continued*

G522N, Cin8$_{NL}$Cut7 and Cin8$_{NL}$Eg5-NG) and diffusive Cin8-3GFP localization in the nucleus and in association with nuclear MTs (bottom, as in Cin8$_{NL}$Eg5 and Cin8$_{NL}$Cut7-NG). (E) Average area (± SEM) of Cin8-3GFP localization in the nucleus of each variant, indicated on the x-axis, was calculated by the particle analysis function in ImageJ software (see Materials and methods and *Source data 3*). Numbers of examined cells for each variant are indicated in the graph columns. **p < 0.01, compared to wt Cin8.

The online version of this article includes the following source data for figure 2:

**Source data 1.** Cell cycle analysis of cells expressing NL variants of Cin8.

**Source data 2.** Area of localization of NL variants of Cin8 at the poles in cells with monopolar spindles.

---

indicating that localization of this variant is significantly more diffusive in the nucleus, probably due to reduced affinity to nuclear MTs. In addition to the diffusive nuclear localization and in contrast to wt Cin8 and Cin8$_{NL}$Cut7, Cin8$_{NL}$Eg5 also exhibited residual attachment to the nuclear MTs (*Figure 2C*, white arrow). Such localization, similar to the previously reported pattern of the tailless Cin8 mutant that had lost its minus-end directionality preference (*Düselder et al., 2015*; *Shapira et al., 2016*) suggests that the minus-end directed motility is impaired in the Cin8$_{NL}$Eg5 variant, although it is maintained in the partially functional Cin8$_{NL}$Cut7. This is probably the reason for the ability of Cin8$_{NL}$Cut7, but not Cin8$_{NL}$Eg5, to support yeast viability.

In summary, the above experiments indicate that replacement of the Cin8 NL sequence with the sequence from plus-end directed kinesin motors produced variants that are not functional in cells, probably due to decreased affinity to MTs and abolished minus-end directed motility on nuclear MTs prior to spindle assembly. In contrast, replacement of the Cin8 NL sequence with a sequence from the bi-directional Cut7 probably maintained minus-end directed motility, resulting in a partially functional variant of Cin8 that can support cell viability as a sole source of kinesin-5.

## Importance of glycine in the N-latch position for the intracellular function of Cin8

Cin8 contains glycine at position 522 in the NL, which is conserved in kinesin-5 homologs of the *Saccharomycetes* class (*Figure 1—figure supplement 1*). In contrast, the majority of other kinesin motors contain asparagine in this position (*Figure 1B*), which is predicted to serve as a latch (N-latch), stabilizing the docked NL along the core motor domain via core H-bonds to a conserved glycine in a loop between α1 and β3 of the motor domain and to additional amino acids in β7 (*Hwang et al., 2008*). We found that the variant in which this glycine had been replaced with asparagine, namely, Cin8-G522N, exhibited reduced viability at 35°C and 37°C, when expressed as the sole source of kinesin-5 (*Figure 1C*). Consistently, *cin8Δ kip1Δ* cells expressing the Cin8-G522N variant exhibited longer doubling times than cells expressing wt Cin8 (*Table 1*), thereby indicating that the functionality of Cin8 was undermined by the replacement of glycine at position 522 with asparagine. To examine the role of this glycine, we generated a mutant of Cin8 containing the NL sequence of the plus-end directed Eg5, with the asparagine at position 522 mutated back to glycine; this mutant is designated Cin8$_{NL}$Eg5-NG. Strikingly, reinstating the original glycine rescued the non-viable phenotype of the Cin8$_{NL}$Eg5 variant, supporting the viability of *cin8Δ kip1Δ* cells at temperatures up to 30°C (*Figure 1C*). The doubling time of *cin8Δ kip1Δ* cells expressing Cin8$_{NL}$Eg5-NG was longer than that of cells expressing wt Cin8 (*Table 1*). However, in the presence of Kip1, and in contrast to cells expressing Cin8$_{NL}$Eg5, no accumulation of monopolar spindles was observed in cells expressing the Cin8$_{NL}$Eg5-NG variant (*Figure 2A and B*). Finally, prior to spindle assembly, the Cin8$_{NL}$Eg5-NG variant localized near the SPBs, at the minus-end of the nuclear MTs, similarly to the localization pattern of wt Cin8 (*Figure 2C–E*). This pattern suggests that, in contrast to Cin8$_{NL}$Eg5, minus-end directed motility is restored in the Cin8$_{NL}$Eg5-NG variant, which moves in the minus-end direction on nuclear MTs and concentrates at the SPBs (*Shapira et al., 2017*). Taken together, these results indicate that the presence of glycine at position 522, as in the original sequence of Cin8, rescued the majority of defects of the non-functional Cin8$_{NL}$Eg5 variant.

To examine the generality of our findings, we investigated whether changing the N-latch asparagine back to a glycine could also rescue the partial viability defect in cells expressing the Cin8$_{NL}$Cut7 variant. Surprisingly, we found that whereas cells expressing the Cin8$_{NL}$Cut7 variant were partially

viable at 23˚C and 26˚C, cells expressing the variant in which the N-latch asparagine had been replaced by glycine (termed Cin8$_{NL}$Cut7-NG) were not viable at all the examined temperatures (*Figure 1C*). Consistently, cells expressing the Cin8$_{NL}$Cut7-NG variant accumulated with monopolar spindles (*Figure 2A*), indicating that this variant is defective in the spindle assembly function. Finally, the Cin8$_{NL}$Cut7-NG variant exhibited diffusive localization in higher percent of cells with monopolar spindles, compared to wt Cin8 (*Figure 2C–E*). These results indicate that in contrast to the Cin8$_{NL}$Eg5 scenario, when the NL of Cin8 is replaced by the NL from the bi-directional Cut7, reinstating the original glycine of Cin8 resulted in a non-functional Cin8 variant that did not rescue the viability defects of Cin8$_{NL}$Cut7, but rather exacerbated these defects.

## NL variants of Cin8 exhibit a different H-bond arrangement in the docked orientation

To seek an explanation for the differences in activity of the NL variants of Cin8, and in particular, for the finding that replacing the N-latch asparagine with glycine rescued the defects of the Cin8$_{NL}$Eg5 variant, but not the Cin8$_{NL}$Cut7 variant, we examined the structural configuration of the NL docking by using homology modeling. To this end, we generated structural models of the variants in a nucleotide-bound state in which the NL is docked by exploiting four known structures of kinesin motors published previously, all in the presence of adenylyl-imidodiphosphate (AMP-PNP) (*Figure 3* and *Figure 3—figure supplement 1*, and *Table 2*) (see Materials and methods). In the modeling, we focused on the H-bond array between the NL and the motor domain (*Figure 3* and *Figure 3—figure supplement 1*, dashed lines). In this context, we note that it has previously been suggested that the non-motor N-terminus of the plus-end directed kinesin-5 Eg5 forms a β-sheet cover strand bundle (CSB) with β9 of the docked NL, which stabilizes the docked NL configuration (*Goulet et al., 2012*; *Goulet et al., 2014*), similarly to reports for kinesin-1 motors (*Hwang et al., 2008*; *Khalil et al., 2008*). A recent cryo-EM study of MT-bound *S. pombe* Cut7 indicates that a similar CSB formation may also take place during NL docking in bi-directional kinesin-5 motors (*von Loeffelholz et al., 2019*), despite the fact that the non-motor N-terminal region is considerably longer in bi-directional kinesin-5 motors compared to exclusively plus-end directed kinesin-5 and kinesin-1 motors (*Singh et al., 2018*). Thus, it is likely that CSB formation stabilizes NL docking in Cin8 and may affect NL dynamics. Indeed, our modeling revealed that β9 of the NLs of all our Cin8 variants forms two H-bonds with the non-motor N-terminal residues, L73 and I75 (*Figure 3—figure supplement 1*, red dashed lines), supporting the notion that the N-terminal CS of Cin8 stabilizes the docked conformation of the NL. However, since a large part of the N-terminal non-motor sequence is absent in our models (due to a lack of structural data for this region), at this stage we cannot estimate accurately to what extent these extended non-motor N-terminal sequences contribute to the stabilization of NL docking. Therefore, the other H-bonds between β9 and the motor domain present in our models (*Figure 3—figure supplement 1*, black dashed lines) are likely to be less informative. In addition, the formation of H-bonds in this region does not correlate with the function of Cin8. For example, wt Cin8 and Cin8$_{NL}$Cut7 exhibit drastically different viability in cells (*Figure 1C*), but they form the same number of H-bonds between β9 and the motor domain (*Figure 3—figure supplement 1*, black dashed lines). Thus, we conclude that the differences in H-bond formation between β9 and the motor domain, as depicted by our modeling, cannot explain the differences in functionality between the NL variants of Cin8.

We next examined the H-bond formation between the N-latch position and β10 of the NL and the motor domain. According to our models, wt Cin8 forms a conserved backbone H-bond between G157 in a loop between α1 and β3 and the N-latch G522 in the NL (*Hwang et al., 2008*). In addition, wt Cin8 forms two H-bonds between F412 in β7 and S523 and F524 in the NL and an additional H-bond between S523 and F524 of the NL (*Figure 3* and *Table 2*). This array of H-bonds was not recapitulated in any of the variants we examined, which might explain why none of these variants is able to exhibit the full function of Cin8 in cells (*Figures 1* and *2* and *Table 1*). The two partially functional variants, Cin8-G522N and Cin8$_{NL}$Eg5-NG, also form the conserved backbone H-bond between G522/N522 of the NL and G157 in a loop between α1 and β3. However, both these variants form only one H-bond between F245/K245 in the NL and F412 in β7. Finally, the Cin8$_{NL}$Cut variant, which exhibits severely impaired activity in cells (*Figures 1* and *2* and *Table 1*), forms only one backbone H-bond between N522 of the NL and G157 and completely lacks a H-bond between the NL and β7

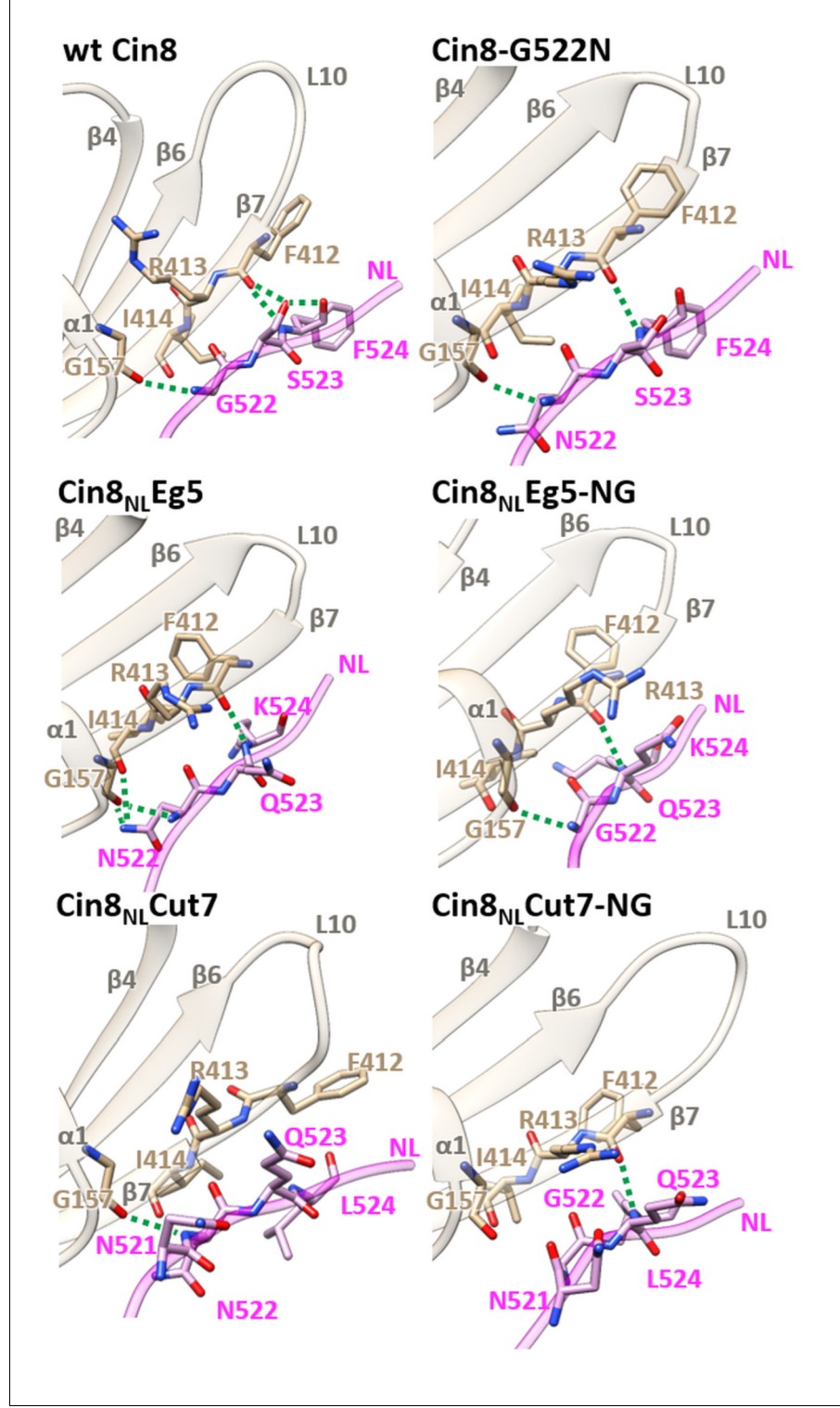

**Figure 3.** Structural analysis of NL/β7 in different variants of Cin8. 3D homology models of Cin8 NL variants were generated on the basis of four PDB structures of kinesin motors published previously, all in the presence of AMP-PNP. Structural elements of the motor domain (gray) and NL (magenta) are depicted in ribbon representation; residue elements such as oxygen and nitrogen are shown in red and blue respectively; H-bonds formed between NL residues (magenta) and motor-domain residues (tan) according to the calculated donor-acceptor distances and the donor-acceptor-hydrogen angles deduced from the models are shown as green dashed lines.

The online version of this article includes the following figure supplement(s) for figure 3:

*Figure 3 continued on next page*

*Figure 3 continued*

**Figure supplement 1.** Analysis of H-bond interactions between β9 of the NL and the motor domain of Cin8.

(*Figure 3* and *Table 2*). These results suggest that H-bond stabilization between NL and β7 is important, but not critical, for the function of Cin8.

Interestingly, the non-functional Cin8$_{NL}$Eg5 variant also forms only one H-bond between F412 in β7 and K524 in the NL. However, according to our models, it forms a highly stabilized H-bond array involving the N-latch asparagine (N522) of the NL, which replaces the G522 of Cin8. In Cin8$_{NL}$Eg5, similarly to wt Cin8, a conserved backbone H-bond is formed between G157 and N522. In addition, two H-bonds are formed between the amide nitrogen of N255, one with G157 and the other with I414 in β7 (*Figure 3* and *Table 2*). Similarly to Cin8$_{NL}$Eg5, the partially functional Cin8$_{NL}$Eg5-NG variant also forms only one H-bond between F412 in β7 and K524 in the NL. However, in contrast to Cin8$_{NL}$Eg5, it does not form the two stabilizing H-bonds between the amide nitrogen of N522, and G157 and I414. Since the Cin8$_{NL}$Eg5-NG variant is partially functional in cells, while Cin8$_{NL}$Eg5 is not (*Figures 1* and *2* and *Table 1*), we propose that the extra stabilization of the NL during docking, provided by H-bonds of the amide group of N522, is one of the reasons for lack of functionality of Cin8$_{NL}$Eg5 in cells. Finally, the non-functional Cin8$_{NL}$Cut7-NG forms only one H-bond between F412 in β7 and L524 of the NL and is unable to form the highly conserved H-bond between G157 and G522 (*Figure 3* and *Table 2*). This might lead to significant decrease of stabilization of the NL docking and to Cin8$_{NL}$Cut7-NG to failing to perform essential functions in cells. Taken together, our data indicate that NL docking to the Cin8 core motor domain is governed by H-bonds formed in two regions, one between G522 in the N-latch position and G157 and the other between F412 in β7 and residues in positions 523 and 254 of the NL. Of these two regions, the exact stabilization of the H-bond formation at the N-latch position has a more significant impact on the function of Cin8 in cells.

## Configuration of the H-bond during NL docking regulates the motile properties of Cin8 in vitro

To correlate between the intracellular phenotypes of the NL variants and their motor functions, we examined their activity in vitro. In these assays, we characterized the activity of GFP-tagged full-

**Table 2.** H-bond array in the modeled 3D structures of Cin8 variants between the N-latch position and β10 of the NL and motor domain.

| Protein | Residue i | Atom | Residue J | Atom |
|---|---|---|---|---|
| wt Cin8 | F412 | O | F524 | N |
| | F412 | O | S523 | Oγ |
| | F524 | N | S523 | Oγ |
| | G157 | O | G522 | N |
| Cin8$_{NL}$Eg5 | F412 | O | K524 | N |
| | G157 | O | N522 | Nδ |
| | G157 | O | N522 | N |
| | I414 | O | N522 | Nδ |
| Cin8-G522N | F412 | O | F524 | N |
| | G157 | O | N522 | N |
| Cin8$_{NL}$Eg5-NG | F412 | O | K524 | N |
| | G157 | O | G522 | N |
| Cin8$_{NL}$Cut7 | G157 | O | N522 | N |
| Cin8$_{NL}$Cut7-NG | F412 | O | L524 | N |

H-bonds are based on geometric criteria. Here, we used stringent criteria: the distance between the donor and acceptor heavy atoms must be <0.36 nm, and the acceptor-donor hydrogen angle must be <30°.

length NL variants, overexpressed and purified from *S. cerevisiae* cells, on fluorescently labeled GMPCPP-stabilized MTs (*Pandey et al., 2021*; *Shapira et al., 2017*). We first compared the MT affinities of the NL variants by determining the average number of MT-attached motors per MT length at equal motor protein concentrations. We found that all the NL variants exhibited significantly lower levels of MT-bound motors than wt Cin8 (*Figure 4A*), thereby indicating that the NL affects the MT-affinity of Cin8. Moreover, we also found that the majority of Cin8$_{NL}$Eg5 and Cin8$_{NL}$-Cut7-NG variants, which were unable to support yeast cell viability as a sole source for kinesin-5, exhibited no motor motility at 140 mM KCl (not shown). We also examined the motility of Cin8$_{NL}$Eg5 at a lower salt concentration, at which the affinity of the motors to MTs is increased (*Shapira and Gheber, 2016*). We found that under these conditions, Cin8$_{NL}$Eg5 exhibited only bi-directional diffusive motility, with no minus-end directed bias, in contrast to wt Cin8, which exhibited processive minus-end directed motility (*Figure 4B*). These results indicate that, consistent with its intracellular localization (*Figure 2C–E*), the Cin8$_{NL}$Eg5 variant exhibited very low MT affinity and was unable to move towards the minus-end of the MTs.

We then examined the motility of the *functional* Cin8 NL variants at a saturating ATP concentration and at a high ionic strength (140 mM KCl). We found that under these conditions, consistent with previous reports (*Gerson-Gurwitz et al., 2011*; *Shapira et al., 2016*; *Shapira et al., 2017*), all Cin8 NL variants moved processively in the minus-end direction of the MTs (*Figure 4C*). We have previously demonstrated that one of the factors that affects the directionality and velocity of Cin8 is its accumulation in clusters on MTs (*Shapira et al., 2017*). Thus, to examine the effects of mutations in the NL sequence on motility but not on motor clustering, we sorted out single molecules of Cin8 from a total population of moving Cin8 particles, based on their fluorescence intensities (*Pandey et al., 2021*) (see Materials and methods). By following the fluorescence intensity of Cin8 particles as a function of time, we observed single events of intensity decrease, of ~45 arbitrary intensity units (a.u.), most probably originating from the photobleaching of single GFP molecules (*Figure 4—figure supplement 1A*). Since single Cin8 motors are tetramers comprised of four identical subunits (*Hildebrandt et al., 2006*), the maximal fluorescence intensity of a single Cin8 molecule containing four GFP molecules is expected to be ~180 a.u. The intensity distribution of the total Cin8-GFP population was consistent with this notion, in that it exhibited a major intensity peak containing ~65% of Cin8 particles with intensity <180 a.u. (*Figure 4—figure supplement 1B*). The maximal intensity of the peak was ~120 a.u., consistent with the average intensity of single Cin8 molecules containing one, two, three, or four fluorescent GFP molecules (*Figure 4—figure supplement 1B*). Based on this analysis (see Materials and methods), we defined 'single Cin8 molecules' as particles of intensity lower than 180 a.u. and analyzed the motile properties only of those Cin8 molecules, thereby ensuring analysis of mainly single molecules of tetrameric Cin8 NL-variants (*Pandey et al., 2021*).

We quantified the motility of single molecules of NL variants by tracing their positions on the MTs at each time point, followed by mean displacement (MD) analysis (see Materials and methods). Consistent with previous reports (*Gerson-Gurwitz et al., 2011*; *Roostalu et al., 2011*; *Shapira and Gheber, 2016*; *Shapira et al., 2017*), we found that under high ionic strength conditions, single molecules of wt Cin8 exhibited fast processive movements towards the minus ends of MTs, with a high average velocity of −318±4 nm/s (*Figure 4C and D*). Remarkably, in contrast to the diffusive motility of the Cin8 variant containing the NL sequence of the plus-end directed Eg5 (Cin8$_{NL}$Eg5) (*Figure 4B*, orange arrows), mutating asparagine at position 522 to glycine (Cin8$_{NL}$Eg5-NG) dramatically changed the motile behavior, inducing processive minus-end directed motility (*Figure 4B–D*). Although the average velocity of the Cin8$_{NL}$Eg5-NG variant was lower than that of wt Cin8 (*Figure 4D*), the processive minus-end directed motility of this variant under the high ionic strength conditions indicates that glycine at position 522 in the NL modulates the minus-end directed motility of Cin8. Consistently, in wt Cin8, mutation of glycine in this position to asparagine resulted in slower minus-end directed motility, compared with that of the original wt Cin8 (*Figure 4D*). Furthermore, Cin8$_{NL}$Cut7, the Cin8 variant with the NL sequence of the bi-directional Cut7 containing asparagine at position 522 (*Figure 1C*), also exhibited processive minus-end directed motility, but with reduced velocity compared with that of wt Cin8 (*Figure 4D*).

Finally, we also examined the run length of motile trajectories of single NL variants (*Figure 4—figure supplement 2*). We found that the average run length of the motile Cin8$_{NL}$Cut7 was significantly shorter than that of wt Cin8. Among the viable NL variants, Cin8$_{NL}$Cut7 exhibited the lowest

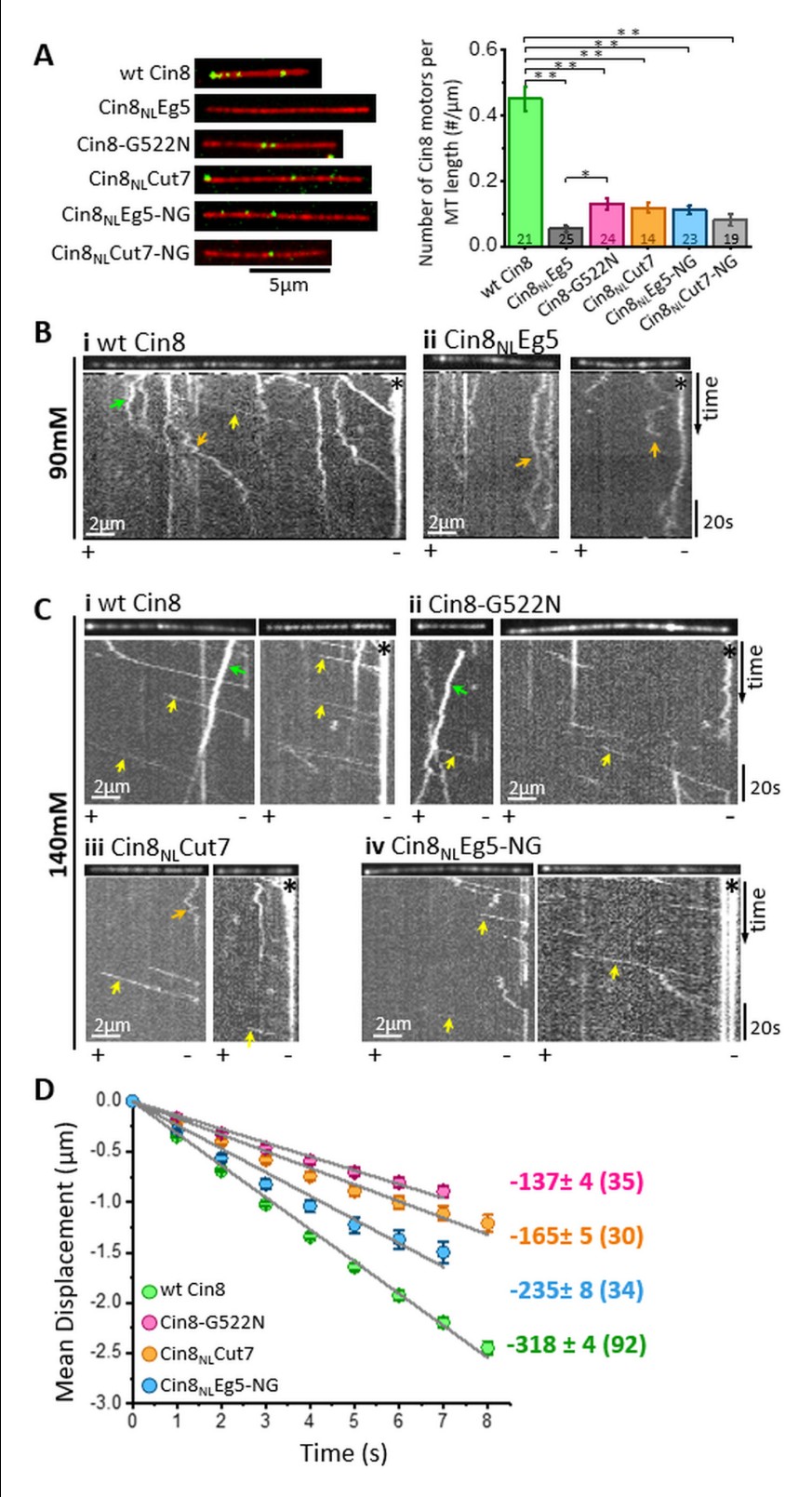

**Figure 4.** In vitro MT binding and single molecule motility assay of NL variants of Cin8. (A) MT-binding assay of GFP-tagged NL Cin8 variants in the presence of 1 mM ATP and 140 mM KCl. Left: Representative images of motors (green) bound to fluorescently labeled MTs (red) of the variants; bar: 5 µm. Right: Average number (± SEM) of Cin8 motors per MT length. The total number of MT-bound Cin8 motors was divided by the total MT length

*Figure 4 continued on next page*

*Figure 4 continued*

and averaged over 14–25 observation areas, indicated in the graph columns for each NL variant of 346 μm$^2$ (see Materials and methods and *Source data 4*); NL variants are indicated on the X-axis. *p < 0.05; **p < 0.005. (**B, C**) Representative kymographs of single molecule motility assay of NL variants at (**B**) 90 mM KCl and (**C**) 140 mM KCl. The MTs are shown on the top of the kymographs. The directionality of the MTs, indicated at the bottom of each kymograph, was assigned according to the bright plus-end label and/or by the directionality of fast Cin8 minus-end directed movements (*Gerson-Gurwitz et al., 2011*; *Shapira et al., 2017*). Yellow, orange, and green arrows indicate fast minus-end directed, bi-directional, and plus-end directed movements, respectively; asterisks indicate Cin8 clustering at the minus-end of MTs. (**D**) Plots of mean displacement (MD) (± SEM) of single molecules of Cin8 NL variants as a function of time. The solid lines represent linear fits of the mean displacement (*MD = v.t*, where *v* is the velocity, and *t* is time). Numbers on the right indicate mean velocity (nm/s ± SD), calculated from the linear fits of the MD plots as a function of time. Numbers of analyzed trajectories are indicated in parentheses (see Materials and methods and *Source data 5*).

The online version of this article includes the following source data and figure supplement(s) for figure 4:

**Source data 1.** Affinity of NL variants of Cin8 to the MTs.
**Source data 2.** MD analysis of NL variants of Cin8.
**Figure supplement 1.** Fluorescence intensity bleaching and distribution, of Cin8-GFP molecules.
**Figure supplement 2.** Motility trajectories and run length of NL variants.

---

ability to support cell viability as a sole source of kinesin-5 (*Figure 1C*). Thus, our results suggest that that the ability to produce processive minus-end directed motility is one of the motility traits that are important for the intracellular functioning of Cin8.

## The NL of Cin8 regulates its MT-crosslinking

The intracellular functions of kinesin-5 motors are attributed to their ability to crosslink the interpolar MTs of the spindle (*Goulet and Moores, 2013*; *Mann and Wadsworth, 2019*; *Singh et al., 2018*). Thus, to establish additional correlations between intracellular phenotypes and the motor activity of the NL variants, we examined their MT-crosslinking activity in vitro. In this assay, purified Cin8 variants were mixed with fluorescently labeled GMPCPP-stabilized MTs in solution, followed by imaging and quantitation of the fluorescence intensity of the MTs (see Materials and methods). Consistent with previous reports (*Gheber et al., 1999*), we found that Cin8 induced the accumulation of MT bundles, thereby increasing the apparent fluorescence intensity of these bundles compared with that of single MTs (*Figure 5*). The MT-bundling activity of wt Cin8 was concentration dependent, inducing MT bundling at a minimal molar ratio of ~1:40 Cin8 motors to tubulin dimer (*Figure 5D*).

With the exception of Cin8$_{NL}$Eg5, all NL variants (*Figure 1C*) induced MT bundling in a concentration-dependent manner, with the fluorescence intensity of bundles induced by high concentrations of motors being significantly higher than that of MTs without the addition of motors (*Figure 5D*). Moreover, at high motor protein concentrations, the functional NL variants, Cin8-G522N, Cin8$_{NL}$Cut7 and Cin8$_{NL}$Eg5-NG, induced MT bundling to a significantly greater extent compared with the non-functional Cin8$_{NL}$Eg5. This finding indicates that the MT-bundling activity of the NL variants is strongly correlated with the ability of these variants to support cell viability as a sole source of kinesin-5 (*Figure 1C*). However, all the *functional* NL variants exhibited reduced ability to bundle MTs, compared with wt Cin8, a finding that is consistent with the reduced affinity for MTs of these variants (*Figure 4A*). The Cin8-G255N variant exhibited reduced bundling ability at low and intermediate protein concentrations but reached the MT-bundling levels of wt Cin8 at high protein concentrations. In contrast, both the Cin8$_{NL}$Cut7 and Cin8$_{NL}$Eg5-NG variants exhibited reduced MT bundling compared with wt Cin8 at low and high motor concentrations, with Cin8$_{NL}$Cut7 exhibiting better MT-bundling activity than Cin8$_{NL}$Eg5-NG (*Figure 5D*). Finally, in contrast to the Cin8$_{NL}$Eg5 variant, which failed to induce MT bundling, a single change of the asparagine at position 522 of Cin8$_{NL}$Eg5 to glycine, produced a variant that induced MT bundling at high protein concentrations (*Figure 5D*), thereby indicating that glycine at position 522 is important for the MT-crosslinking and bundling activity of Cin8.

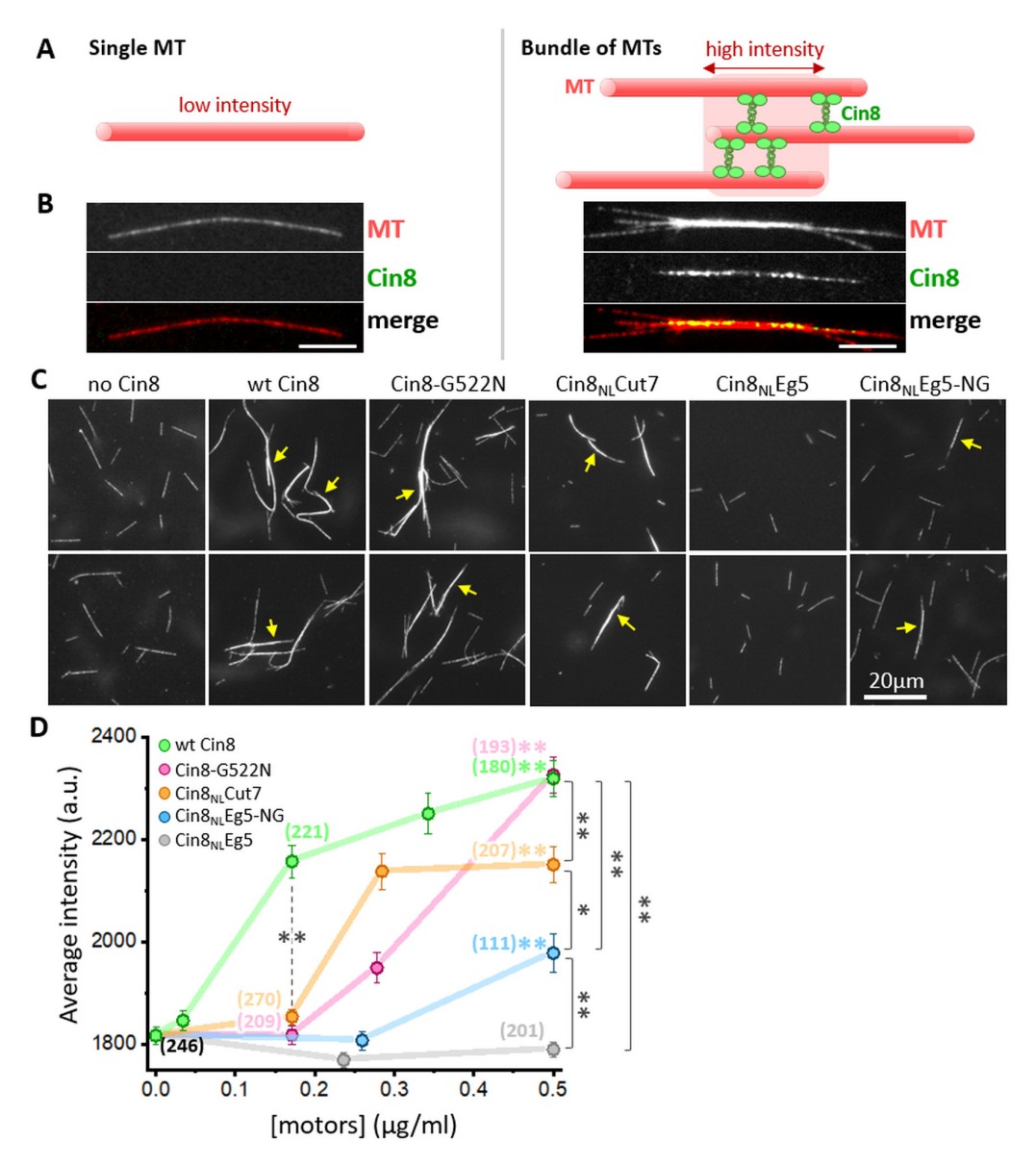

**Figure 5.** In vitro MT bundling by NL variants of Cin8.  (**A**) Schematic representation of MTs (red) cross-linked by Cin8 (green). The left panel represents a single MT with a low fluorescence intensity; the right panel represents a high-fluorescence intensity MT-bundle induced by Cin8. (**B**) Representative images of rhodamine-labeled GMPCPP-stabilized MTs (red) in the absence (left) and presence (right) of Cin8-GFP (green). The MT-bundle presented on the right was induced by wt Cin8-GFP, which was co-localized with the bright section of the MT-bundle. (**C**) Representative images of MT-bundles induced by NL variants of Cin8, indicated on the top. Arrows indicate the bright MT bundles. (**D**) Average intensity (± SEM) of MT bundles as a function of the concentration of Cin8 variants, measured by particle analysis using ImageJ (see Materials and methods *Source data 7*). Numbers of particles analyzed for each variant are indicated in parentheses; black: MTs only without motors. Color-coded asterisks for each NL variant indicate comparison to average MT intensity in the absence of motors. *p < 0.05; **p < 0.005.

The online version of this article includes the following source data for figure 5:

**Source data 1.** MT bundling by NL variants of Cin8.

## Mutations in the NL reduce the efficiency of plus-end directed antiparallel MT sliding

We have previously suggested that for fungal kinesin-5 motors to perform their spindle assembly function in closed mitosis, they need first to localize near the SPBs by their minus-end directed motility and then to reverse directionality to plus-end directed motility between antiparallel MTs in order

to produce antiparallel MT sliding and separate the SPBs apart (*Shapira et al., 2017*). We therefore examined the ability of the NL variants to reverse directionality in an in vitro MT sliding assay, in which one set of MTs was immobilized to a glass surface, followed by the addition of Cin8 with an additional set of MTs (*Figure 6*) (see Materials and methods). By using polarity-marked MTs, both velocity and directionality of MT sliding could be determined for each NL variant. Consistently with previous results (*Shapira et al., 2017*), we found that in the presence of 125 mM KCl and ~3 ng/ml wt Cin8, MT sliding was observed. In more than 50% of cases, this sliding was plus-end directed, with the minus-end of the moving MT leading (*Figure 6* and *Figure 6—video 1*). Under the same conditions, MT capturing or sliding was not observed in the presence of the NL variants, consistent with the lower affinity of the NL variants for the MTs (*Figure 4A*) and their reduced ability to bundle MTs (*Figure 5*). To increase the occurrence of MT sliding, we lowered the KCl concentration to 100 mM and increased the motor concentration until MT sliding was observed (*Figure 6B* and *Figure 6— videos 2–4*). Similarly to wt Cin8, in more than 50% of the cases, sliding produced by the NL variants was plus-end directed (*Figure 6B*), indicating that all the viable NL variants can reverse directionality. The velocity of plus-end MT sliding induced by Cin8-G522N was faster than that of wt Cin8 (*Figure 6B* and *Figure 6—video 2*). The significance of this finding requires further investigation (which we are currently undertaking). Finally, the velocity of plus-end directed MT sliding of the Cin8$_{NL}$Cut7 and Cin8$_{NL}$Eg5-NG variants was significantly slower than that of wt Cin8 (*Figure 6B* and *Figure 6—videos 3* and *4*). Taken together, our data indicate that although the NL variants can reverse to plus-end directionality during antiparallel MT sliding, their MT sliding is less efficient and requires higher motor concentrations compared to wt Cin8.

For all the examined variants, including wt Cin8, we observed bi-directional and minus-end directed MT sliding in addition to the plus-end directed MT sliding discussed above (*Figure 6*, *Figure 6—figure supplement 1* and *Figure 6—videos 5* and *6*). The occurrence of such events was similar for the different variants (*Figure 6B*). We have recently suggested that Cin8 can reverse directionality from minus- to plus-end directed motility due to forces opposing the directional motility, which we referred to as drag (*Pandey et al., 2021*). Such forces can result, for example, from numerous motors interacting with the same crosslinked MTs during antiparallel MT sliding. In accordance with this notion, in the absence of a considerable applied drag force, MT sliding can also be minus-end directed, similarly to the single molecule motility under the high salt conditions (*Figure 4B,C*). Based on this model, we propose that the bi-directional and minus-end directed MT sliding we observed here results from a low local concentration of Cin8 motors in the overlapping MT region and/or reduced MT affinity of the NL variants (*Figure 4A*). However, the fact that all the examined Cin8 motors can produce plus-end directed MT sliding suggests that if sufficient motors are present locally between overlapping MT of the spindle, antiparallel plus-end directed MT sliding can take place to produce an outwardly directed force that separates the SPBs apart.

## Discussion

The role of the NL dynamics in regulating the motor functions of plus-end directed kinesins has been addressed in a number of studies. However, the current study is the first to demonstrate that mutations in the NL modulate the motile properties and intracellular functions of a bi-directional N-terminal kinesin motor. The data presented here indicates that mutations in the NL regulate MT affinity, minus-end directed motility, and MT crosslinking and antiparallel sliding functions of the bi-directional kinesin-5, Cin8. In turn, these motor functions affect the ability of Cin8 to localize at the spindle poles prior to spindle assembly, to form mitotic spindles, and to support cell viability as a sole source of kinesin-5.

### Minus-end directed motility is important for the intracellular function of a bi-directional kinesin-5

The minus-end directed and switchable directionality of fungal kinesin-5 motors was discovered nearly a decade ago. Although a recent study, based on theoretical simulations, suggests that the minus-end directed motility of the bi-directional *S. pombe* kinesin-5 is necessary for spindle assembly (*Blackwell et al., 2017*), experimental support for this notion is still missing. We have recently proposed that in fungal cells, which divide via closed mitosis, minus-end motility of kinesin-5 motors is needed to localize these motors near the spindle poles prior to spindle assembly. At this location,

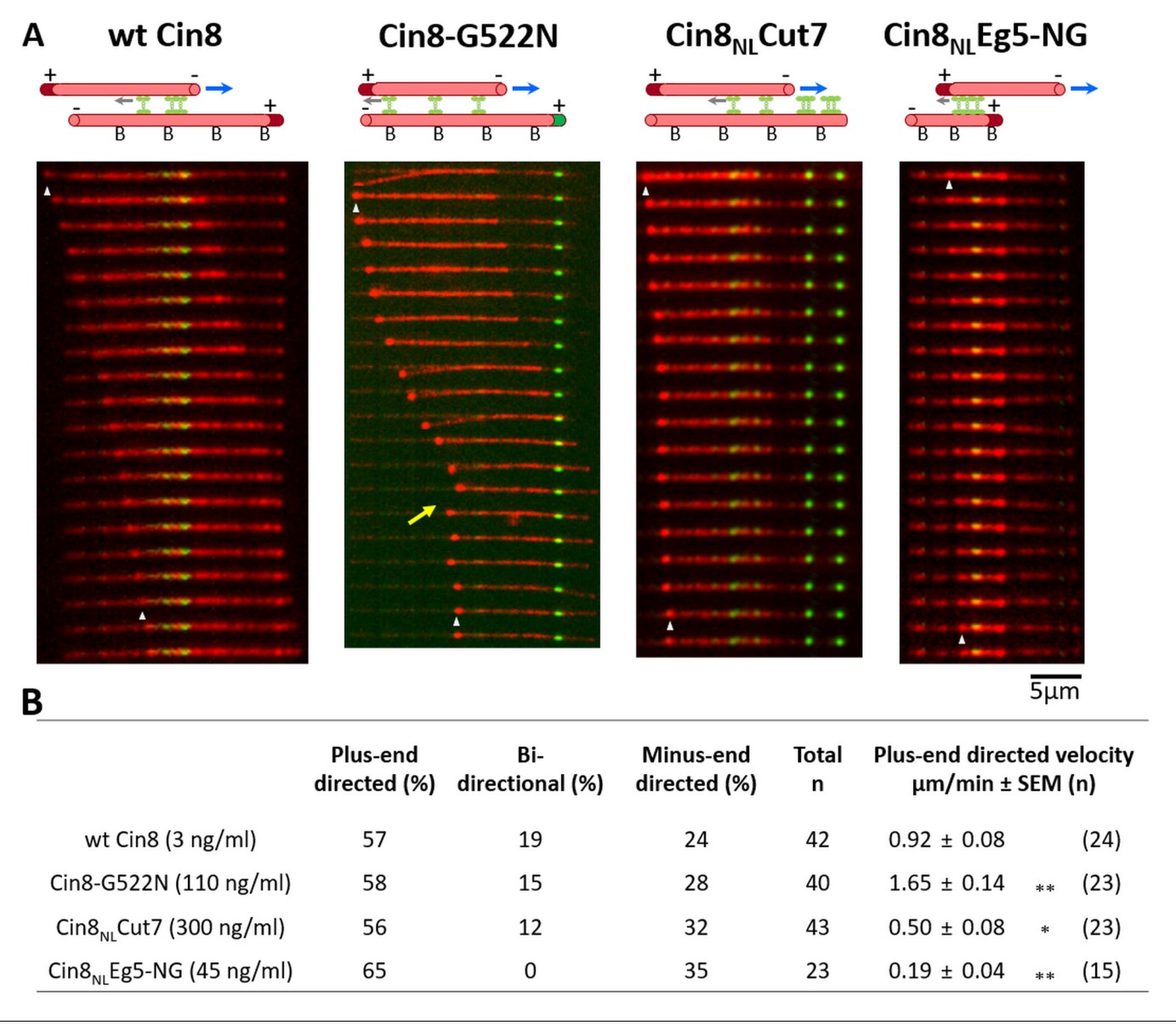

**Figure 6.** Cin8 induced MT sliding of NL variants. (A) Representative time-lapse of a plus-end directed MT sliding event for each NL variant. Yellow arrow indicates a bi-directional movement of the MT during Cin8-induced sliding. In the schematic representations of the first frame of the MT sliding event presented on the top of the panel; rhodamine-labeled MTs are shown as red tubes; plus-end labeling is indicated by dark red or green coloring of the MT; surface binding of the MTs via an avidin-biotin bond is indicated by 'B'; GFP-labeled Cin8 motors are shown as green vertical dumbbell shapes; MT polarities are indicated (for Cin8$_{NL}$Cut7, the stationary MT lacks a polarity label); gray arrows indicate motor directionality; and blue arrows indicate the directionality of moving MT. (B) Characteristic MT sliding induced by the NL variants. Motor concentrations are indicated in parentheses on the left. Mean velocities of plus-end directed MT sliding (± SEM (n)) are indicated on the right, see materials and methods and *Source data 8* for definition. Statistical analysis performed by Dunnett's test for comparing plus-end directed movements of the variants compared to wt Cin8; *p < 0.05, **p < 0.01.

The online version of this article includes the following video, source data, and figure supplement(s) for figure 6:

**Source data 1.** MT sliding by NL variants of Cin8.

**Figure supplement 1.** Examples of bi-directional and minus-end directed MT sliding.

**Figure 6—video 1.** Plus-end directed MT sliding induced by GFP-tagged wt Cin8, presented in *Figure 6A*.
https://elifesciences.org/articles/71036#fig6video1

**Figure 6—video 2.** Plus-end directed MT sliding induced by GFP-tagged Cin8-G522N, presented in *Figure 6A*.
https://elifesciences.org/articles/71036#fig6video2

**Figure 6—video 3.** Plus-end directed MT sliding induced by GFP-tagged Cin8$_{NL}$Cut7, presented in *Figure 6A*.

*Figure 6 continued on next page*

*Figure 6 continued*

https://elifesciences.org/articles/71036#fig6video3

**Figure 6—video 4.** Plus-end directed MT sliding induced by GFP-tagged Cin8$_{NL}$Eg5-NG, presented in *Figure 6A*.

https://elifesciences.org/articles/71036#fig6video4

**Figure 6—video 5.** Bi-directional sliding event induced by GFP-tagged Cin8-G522N, presented in *Figure 6—figure supplement 1*.

https://elifesciences.org/articles/71036#fig6video5

**Figure 6—video 6.** Minus-end directed MT sliding event induced by GFP-tagged Cin8$_{NL}$Cut7, presented in *Figure 6—figure supplement 1*.

https://elifesciences.org/articles/71036#fig6video6

kinesin-5 motors capture and crosslink MTs emanating from the neighboring SPBs and promote SPB separation via antiparallel sliding of the crosslinked MTs (*Shapira et al., 2017*). Analysis of in vivo and in vitro functions of the NL variants of Cin8 presented here support this model. Our data demonstrates that, compared with wt Cin8, all the examined NL variants exhibited reduced affinity for MTs (*Figure 4A*) and an impaired ability to crosslink and slide MTs in vitro (*Figure 5C–D*). Importantly, our data also shows that all NL variants *that can support the viability of yeast cells* exhibit two common traits: (a) they are able to move in the minus-end direction in vitro (*Figure 4*) and (b) they localize near the SPBs prior to spindle assembly (*Figure 2C–E*), indicating a connection between intracellular functionality, minus-end directed motility, and localization at the SPBs. Differences in minus-end directed velocity per se (*Figure 4C and D*) seem to be less important for the intracellular function, as long as the variants can move in the minus-end direction. Thus, we conclude that, consistent with our model (*Shapira et al., 2017*; *Singh et al., 2018*), processive minus-end directed motility that localizes the motors near the SPBs prior to spindle assembly and the subsequent reversal of directionality to produce antiparallel MT sliding are necessary for the mitotic functions of the kinesin-5 Cin8.

## Exact H-bond configuration during NL docking is important for Cin8 functions

Examination by modeling, based on known structures of four different kinesin motors in the AMP-PNP-bound state, reveals that none of the examined NL variants recapitulated the array of H-bonds formed during NL docking in wt Cin8 (*Figure 3* and *Table 2*). All these variants are defective in intracellular functions and/or motile properties (*Figures 1*, *2* and *4–6*), suggesting that an exact H-bond array during NL docking is one of the factors required for their functioning.

According to our modeling, the overall number of H-bonds formed between the N-latch position and β10 of the NL is considerably smaller than that in kinesin-1 motors (*Budaitis et al., 2019*; *Hwang et al., 2008*), suggesting decreased H-bond stabilization and increased flexibility of the docked NL of Cin8. The H-bonds stabilizing Cin8 NL docking in the N-latch–β10 region can be divided into two groups. One is the conserved backbone H-bond between the N-latch glycine G522 and motor domain glycine G157, and other group is comprised of a series of H-bonds between amino acids S523 and F524 in the NL and F412 in β7 of the motor domain (*Figure 3* and *Table 2*). Of these stabilizing H-bond factors, the β7/NL H-bond stabilization seems to be less critical for the function of Cin8, since the Cin8$_{NL}$Cut7 variant that completely lacks H-bonds in this region is able to support cell viability as a sole source of kinesin-5 function, although its intracellular functionality is most defective among the NL variants examined (*Figures 1* and *2* and *Table 1*). The overall functionality of kinesin-5 motors in cells is a complex phenomenon depending on factors such as kinesin-5 motor activity (*Gheber et al., 1999*), expression of and interaction with other spindle proteins (*Khmelinskii et al., 2009*; *Khmelinskii and Schiebel, 2008*) and phospho-regulation of kinesin-5 motors (*Avunie-Masala et al., 2011*; *Goldstein et al., 2019*; *Goldstein et al., 2017*; *Shapira and Gheber, 2016*). Thus, the significantly reduced intracellular functionality of the Cin8$_{NL}$Cut7 variant (*Figure 1C*) can be explained, at least in part, by combination of its defective motor functions, such as reduced MT binding (*Figure 4A*), reduced velocity (*Figure 4D*) and processivity (*Figure 4—figure supplement 2*) on the single-molecule level and reduced efficiency of crosslinking and sliding apart antiparallel MTs (*Figures 5* and *6*).

In contrast to the β7/NL H-bond stabilization, our modeling and experimental data suggest that the conserved backbone H-bond between the N-latch G522 and the motor domain G157 is critical

for Cin8 function, since its absence results in the non-functional Cin8$_{NL}$Cut7-NG variant (*Figure 1C*). Interestingly, it appears that over-stabilization of NL docking by additional H-bonds between N522 and the motor domain is not tolerated by Cin8, since it abrogates the functionality of the 'over-stabilized' Cin8$_{NL}$Eg5 variant (*Figures 1C* and *3*, and *Table 2*); see below.

### Additional H-bond stabilization of the N-latch position during NL docking is incompatible with Cin8 functionality

Although many kinesin motors contain asparagine in the N-latch position equivalent to glycine 522 in Cin8 (*Figure 1B*), our modeling suggests that the conserved backbone H-bond between this glycine and G157 in the motor domain is conserved in Cin8 (*Figure 3* and *Table 2*). However, when the NL of Cin8 is replaced by that of Eg5, additional stabilizing H-bonds are formed between N522 (in the position of G522 of Cin8) and the motor domain in the non-functional variant. In the same variant, when N522 is replaced with glycine, as in Cin8, these additional stabilizing H-bonds are eliminated (*Figure 3* and *Table 2*) and partial functionality in vivo and in vitro is restored (*Figures 1C*, *2* and *4–6*). Thus, the difference between the activity of the Cin8$_{NL}$Eg5 and Cin8$_{NL}$Eg5-NG variants and the differences between their modeled H-bond configuration during NL docking suggest that additional stabilization of the N-latch position by H-bond formation is incompatible with the functioning of Cin8. In other words, certain degree of H-bond flexibility during NL docking must be maintained to enable the functioning of Cin8.

Recent studies indicate that processive stepping in the plus-end direction of kinesin-1 motors is achieved, in part, by a gating mechanism that prevents simultaneous strong binding of the two kinesin-1 motor-domain heads to MTs (*Block, 2007*). It is believed that this gating is carried out by the coupling between the orientation of the NL, the inter-head tension, and ATP binding and hydrolysis, such that ATP binding in the leading head is prevented until the rear head assumes a weakly bound state or detaches (*Clancy et al., 2011*; *Dogan et al., 2015*; *Guydosh and Block, 2006*; *Klumpp et al., 2004*; *Rosenfeld et al., 2001*; *Schief et al., 2004*; *Yildiz et al., 2008*). However, a different gating mechanism probably takes place during motility of plus-end directed kinesin-5 motors (*Muretta et al., 2015*), which allows for simultaneous strong MT-binding of two motor domains of a dimer in a rigor state with no bound nucleotide (*Chen et al., 2016*; *Krzysiak et al., 2008*). The two-headed strongly bound state has been attributed to the longer and more flexible NL of kinesin-5 motors (*Shastry and Hancock, 2010*; *Shastry and Hancock, 2011*; *Yildiz et al., 2008*) and may be necessary for force production during antiparallel sliding of MTs in spindle assembly and elongation (*Gerson-Gurwitz et al., 2009*; *Leary et al., 2019*; *Movshovich et al., 2008*; *Saunders and Hoyt, 1992*; *Sharp et al., 1999*). The bi-directional kinesin-5 motors have the ability to move in both plus-end and minus-end directions, in addition to their ability to crosslink and slide apart antiparallel MTs. Thus, their NL dynamics must be adapted for bi-directional stepping. Since inducing flexibility in the NL by extending the length of the NL was previously shown to enable minus-end directed back-stepping of kinesin-1 dimers (*Clancy et al., 2011*), we propose that some flexibility should be maintained during NL docking of Cin8 to allow bi-directional motility.

### Connection between H-bond stabilization of NL docking and minus-end directed motility of Cin8

In a recent study, the role of NL-docking stabilization in regulating the motility of kinesin-1 was experimentally examined (*Budaitis et al., 2019*). That study reported that mutating the N-latch asparagine in the NL of kinesin-1 (equivalent to glycine 522 in Cin8) to alanine, which eliminates stabilizing interactions, increased the plus-end directed motor velocity in a single-molecule motility assay (*Budaitis et al., 2019*). Interestingly, we observed an opposite effect on the velocity of single-molecules of Cin8 during processive movement in the *minus-end direction*. The velocity of all the viable Cin8 variants was significantly slower in the minus-end direction than that of wt Cin8 (*Figure 4D*). In all these variants, the number of stabilizing H-bonds between the NL and motor domain was reduced compared to that of wt Cin8 (*Figure 3* and *Table 2*). Based on this difference, it is tempting to propose that the NL docked conformation (in the plus-end directed) is one of the intermediate conformations during active stepping of bi-directional kinesin motors. This notion is consistent with recent cryo-EM reports indicating that in the ATP-bound state the NLs of bi-directional kinesin-5 motors assume the docked conformation (*Bell et al., 2017*; *Britto et al., 2016*;

*von Loeffelholz et al., 2019*), similarly to kinesin-1 motors (*Sindelar et al., 2002*; *Sindelar and Downing, 2010*). It is possible that in minus-end directed motility there is a dynamic equilibrium between several orientations of the NL (minus- and plus-end directed). While decreased stabilization of the docked NL in the plus-end direction increases the plus-end directed velocity of an exclusively plus-end directed kinesin (*Budaitis et al., 2019*), it may have an opposite (velocity-slowing) effect on the minus-end directed motility of the bi-directional Cin8.

Finally, although decreased stabilization of NL docking affects the velocity of minus-end directed motility of Cin8 and of plus-end directed motility of kinesin-1 in different ways (*Figure 4D* and *Budaitis et al., 2019*), the intracellular requirements for such stabilization are similar for the two types of motor. For kinesin-1, decreased stabilization of the NL reduced force production and impaired the ability of the mutant motors to transport high-load Golgi cargo vesicles (*Budaitis et al., 2019*). Similarly, the intracellular functions of the bi-directional Cin8, reflected in cell viability, localization to the spindle and doubling times, were defective in cells expressing Cin8 variants with decreased H-bond stabilization of the NL (*Figures 1–3*, and *Tables 1* and *2*). These results suggest that each of the kinesin types is adopted to optimally perform its functions and that different motility characteristics are required for optimal functioning of exclusively plus-end and bi-directional kinesins.

## In summary

we show here that the docking of the NL regulates the motor motility and intracellular functionality of bi-directional kinesin motors. We show that for certain positions of the docked NL (such as the N-latch position), an exact H-bond stabilization is critical for function. At these positions, suboptimal (too strong or too weak) NL docking stabilization is incompatible with kinesin functionality. We believe that this may be a general principle common to all bi-directional kinesin motors (although the exact sequence of the NL can vary). Because of their ability to step in two directions, the bi-directional kinesin motors may be more sensitive to variations of H-bond stabilization at certain positions. Cin8, the bidirectional kinesin investigated in this study, serves as the first example, but further studies on additional bi-directional kinesin motors are needed to examine the generality of the principle demonstrated here.

## Materials and methods

### Molecular cloning and strains

NL variants were generated using standard molecular cloning techniques. All mutations were confirmed by sequencing. Plasmids and *S. cerevisiae* strains used in this study are listed in *Supplementary files 1* and *2*.

### Multiple sequence alignments (MSAs)

MSAs were calculated by the MUSCLE algorithm via Unipro UGENE program (*Edgar, 2004*), color coded by percentage identity with a 50–55% threshold. Sequences of the organisms presented in *Figure 1B* are (from top to bottom): ScCin8 - *Saccharomyces cerevisiae*, Cin8; ScKip1 - *Saccharomyces cerevisiae*, Kip1; SpCu7 - *Schizosaccharomyces pombe*, Cut7; AnBimC - *Aspergillus nidulans*, BimC; kinesin-like protein of *Saccharomyces arboricola*; kinesin-like protein of *Schizosaccharomyces japonicus*; DmKlp61F - *Drosophila melanogaster*, Klp61F; XlEg5 - *Xenopus laevis*, Eg5; human Kif11 – *Homo sapiens* Eg5; DmKHC - *D. melanogaster*, kinesin-1 heavy chain; mouse KHC - *Mus musculus*, kinesin-1 heavy chain; human KHC - *H. sapiens*, kinesin heavy chain isoform 5A.

### Viability assay

The viability of *S. cerevisiae* cells expressing NL variants of Cin8 as the sole source of kinesin-5 was determined as described previously (*Avunie-Masala et al., 2011*; *Düselder et al., 2015*; *Goldstein et al., 2019*; *Shapira et al., 2017*; *Figure 1*). *S. cerevisiae* strains used for this assay were deleted for their chromosomal copies of *CIN8* and *KIP1* and contained an endogenic recessive cyclo-heximide resistance gene (*cin8Δkip1Δcyh$^r$*), containing a plasmid (pMA1208) encoding for wt Cin8 and a wt dominant cycloheximide sensitivity gene (*CYH*); see *Supplementary file 2* for the list of *S. cerevisiae* strains. Following transformation with a plasmid encoding for Cin8 NL variants, the

pMA1208 plasmid was shuffled out by growth on yeast extract-peptone-dextrose (YPD) medium containing 7.5 µg/mL cycloheximide at different temperatures.

## Imaging of *S. cerevisiae* cells

Imaging was performed as previously described (*Goldstein et al., 2019*; *Goldstein et al., 2017*) on *S. cerevisiae* cells deleted of the chromosomal copy of *CIN8*, expressing 3GFP-tagged Cin8 NL variants from its native promoter on a centromeric plasmid, and endogenously expressing a SPB component Spc42 with a tdTomato fluorescent protein. Cells were grown overnight and diluted 2 hr prior to imaging. Z-stacks of yeast cells were acquired using Zeiss Axiovert 200M microscope controlled by the MicroManager software. For spindle length distribution in budded cells, as in *Figure 2A*, projections of the Z-stacks as the distance between the two Spc42-tdTomato SPB components were measured. Cells with monopolar spindles were defined as cells with a small bud and a single Spc42-tdTomato SPB signal. For each variant, three sets of 113–411 cells were categorized according to their SPB morphology and length into three categories: monopolar cells, cells with a short <2 µm spindle, and cells with a long spindle of length > 2 µm. The percentage of budded cells in each category was averaged for three experiments for each of the variants, and statistical analysis was performed by *Dunnett, 1955* compared to wt Cin8 in each spindle length category; degrees of freedom (DF) = 12, and F factors 2.90 ($\alpha$=0.05) and 3.81 ($\alpha$=0.01). Analysis of Cin8 localization in cells with monopolar spindles (*Figure 2C–E*) was performed on 2D projections generated by ImageJ software of monopolar cells. The area of Cin8 localization was measured on images of the fluorescent signal of Cin8-3GFP as follows. First, we generated a mask by thresholding the image using Phansalkar local threshold method of Cin8-3GFP signal in ImageJ. Then, the mask was applied to the original image, resulting an image with a zero background. Finally, the area occupied by the Cin8 signal was measured using the particle analysis function in ImageJ. This analysis was performed for 18–57 cells for each Cin8 NL variant and averaged as indicated in the graph columns in *Figure 2E*. Statistical analysis was performed by *Dunnett, 1955*; DF = 254, and F factors 2.45 ($\alpha$=0.05) and 3.09 ($\alpha$=0.01).

## 3D model structural modeling

Homology modeling of the Cin8 motor domain of wt Cin8 and its NL mutants was performed on the basis of the following four 3D structures that were used as templates, all in the presence of adenylyl imidodiphosphate (AMP-PNP): (a) an X-ray structure of KIF1A kinesin heavy chain isoform 5C (1VFV; *Nitta et al., 2004*); (b) an X-ray structure of Eg5 motor domain (3HQD; *Parke et al., 2010*); (c) an X-ray structure of KIF4A isoform 4A (3ZFC; *Chang et al., 2013*); and (d) a cryo-EM structure of *S. pombe* kinesin-5 Cut7 decorating MTs (6S8M; *von Loeffelholz et al., 2019*). For each variant, 1000 homology models of kinesin-5 Cin8 (residues 73–524 excluding loop 8; residues 297–395) were built using MODELLER v9.25 (*Sali and Blundell, 1993*) and sorted by a Discrete Optimized Protein Energy (DOPE-HR; using a bin size of 0.0125 nm) score. For each modeled residue, the rotamer with the best score was selected by MODELLER v9.25 (*Sali and Blundell, 1993*). No further changes to the rotamers of the residues selected by MODELLER v9.2 were applied. Calculation of H-bonds was based on the distance between donor and acceptor atoms being smaller than 0.35 nm, and the angle between the acceptor, donor and hydrogen atoms being smaller than 30°. Structures were depicted by UCSF Chimera software (*Pettersen et al., 2004*).

## Purification of Cin8-GFP

Overexpression and purification of Cin8-GFP from *S. cerevisiae* cells was performed as previously described (*Shapira and Gheber, 2016*; *Shapira et al., 2017*). Cells expressing Cin8-TEV-GFP-6HIS in a protease-deficient *S. cerevisiae* strain under the GAL1 promoter on a 2µ plasmid were grown in liquid medium supplemented with 2% raffinose. For overexpression, 2% galactose was added for 5 hr. Cells were re-suspended in a lysis/binding buffer (50 mM Tris, 30 mM PIPES, 500 mM NaCl, glycerol 10%, 2 mM $\beta$-mercaptoethanol, 1 mM phenylmethylsulfonyl fluoride, 1 mM MgCl$_2$, 0.1 mM ATP, 0.2% Triton X-100, Complete Protease Inhibitor (Roche), pH 7.5) and snap-frozen in liquid nitrogen. Cell extracts were prepared by manually grinding in liquid nitrogen in lysis/binding buffer. Ni-NTA beads (Invitrogen) were then incubated with the cell extract for 1.5 hr at 4°C, loaded onto a column, and washed with washing buffer (50 mM Tris, 30 mM PIPES, 500 mM NaCl, 30 mM imidazole,

10% glycerol, 1.5 mM β-mercaptoethanol, 0.1 mM Mg-ATP, 0.2% Triton X-100, pH 7.5). Cin8 was eluted with 6 ml of elution buffer (50 mM Tris, 30 mM PIPES, 500 mM NaCl, 250 mM imidazole, 10% glycerol, 1.5 mM β-mercaptoethanol, 0.1 mM Mg-ATP, and 0.2% Triton X-100, pH 7.5). Eluted samples were analyzed by SDS-PAGE and kept frozen in −80°C for further use. Protein concentration of Cin8 motors was estimated by measuring the intensity of the Cin8 band on Coomassie Brilliant Blue stained SDS-PAGE gels, using known concentrations of bovine serum albumin for calibration.

## Single molecule motility assay

The single molecule motility assay was performed on piranha-cleaned salinized coverslips as previously described (*Shapira et al., 2017*): MTs were polymerized by incubating a tubulin mixture containing biotinylated tubulin (T333P), rhodamine-labeled tubulin (TL590M), and unlabeled tubulin (T240) for 1 hr at 37°C with guanosine-5'-[(α,β)-methyleno]triphosphate (GMPCPP). Then, additional rhodamine-labeled tubulin was added to the polymerizing MTs for 1 hr at 37°C to form a bright plus-end labeled cap. Flow chambers with immobilized MTs were prepared as previously described (*Shapira et al., 2017*). Cin8, ~3 ng/ml, in motility buffer (50 mM Tris, 30 mM PIPES, 110–140 mM KCl, 5% glycerol, 2 mM MgCl$_2$, 1 mM EGTA, 30 µg/ml casein, 1 mM DTT, 1 mM ATP, pH 7.2 ATP-regeneration system containing 0.05 mg/mL of phosphocreatine and 0.01 M creatine-kinase) was added to the immobilized MTs and immediately imaged with a Zeiss Axiovert 200M-based microscope, with a 100× objective, equipped with a sCMOS Neo camera. One frame of MTs was captured, followed by time sequence imaging of 90 s with 1 s intervals of Cin8-GFP signal.

## Image and data analysis of the motility assay

Kymographs were generated using ImageJ software for MTs with both ends visible. Directionality of the MTs was assigned on the basis of bright plus-end labeling or by the direction of fast-moving Cin8 particles under high-salt conditions (*Gerson-Gurwitz et al., 2011*; *Shapira et al., 2017*). To distinguish between single Cin8-GFP tetramers and clusters, we followed the intensity of stationary Cin8-GFP particles for 90 s, with intervals of 1 s by using TrackMate plugin in ImageJ. Images were background-subtracted and corrected for uneven illumination (*Pandey et al., 2021*; *Tinevez et al., 2017*). Since GFP stochastically and irreversibly photobleaches over time, we observed single photobleaching steps, probably representing photobleaching of a single GFP molecule (*Figure 4—figure supplement 1A*). Averaging of the intensity of the photobleaching steps over seven observation fields yielded a value of 45±1 (SEM) a.u. (n = 37), representing the intensity contribution of a single GFP molecule. Therefore, Cin8-GFP particles that have the intensity of four GFPs or less are likely to be single Cin8-GFP tetramers (*Pandey et al., 2021*). Mean displacement (MD) analysis was performed as previously described (*Kapitein et al., 2008*; *Shapira and Gheber, 2016*). For MD analysis, only moving particles with the fluorescence intensity of a single Cin8-GFP molecule (< 180 a.u.) were measured. In addition, measures were taken to minimize the effect of GFP photobleaching on the determination of the Cin8 cluster size. We determined the lifetime of a GFP molecule before photobleaching under our experimental conditions to be 23±3 (SEM) s (n = 40). Consequently, based on this estimation, all the motility measurements were performed only on those Cin8 motors that moved within the first 30 s of each measurement. MD analysis was performed by following the position of such particles, either using TrackMate plugin in ImageJ (*Tinevez et al., 2017*) or manually, thereby deducing the displacement at each time interval, followed by averaging the displacement. Average velocity was calculated by fitting the MD analysis to a linear fit corresponding to the equation $MD = v \cdot t$, where $v$ is velocity and $t$ is time.

## Affinity of Cin8 to MTs

The affinity of Cin8 to MTs was measured on stationary GMPCPP-stabilized fluorescently labeled MTs in the presence of 1 mM ATP and 140 mM KCl, as in the single molecule motility assay, keeping the concentration of Cin8-GFP NL variants constant at ~3 ng/ml. One frame of MTs was captured, followed by a one frame of Cin8-GFP signal. Background subtraction was performed by ImageJ software, followed by recognition of Cin8-GFP particles attached to MTs by TrackMate plugin in ImageJ (*Tinevez et al., 2017*) in an observation area of 346 µm$^2$. For each observation area, the number of MT-bound Cin8-GFP particles was divided by the total length of the MTs and averaged for 14–25 observation areas for each Cin8 NL variant, as indicated in the graph columns in *Figure 3A*.

Statistical analysis was performed by one-way ANOVA all pairwise comparison analysis with Tukey correction; DF = 5, and F factor 54.7.

## MT bundling assay

The MT bundling assay was performed by mixing GMPCPP-stabilized fluorescently labeled MTs, as in the single molecule motility assay, with Cin8-GFP NL variants at concentrations ranging from 0 to 0.5 µg/ml in the presence of 140 mM KCl (and the absence of ATP). The mixture was incubated for 10 min at room temperature and imaged by a Zeiss Axiovert 200M-based microscope, as described for the single molecule motility assay. All images were captured under the same illumination conditions and processed by ImageJ, without any manipulation of the images. To calculate the average intensity of MT bundles or single MTs, we first generated a mask by thresholding the image using Phansalkar local threshold method of rhodamine-labeled MTs signal in ImageJ. Then, the mask was applied to the original image, resulting an image with a zero background. Finally, the average intensity of the MTs or MT-bundles was calculated by the particle analysis function in ImageJ, keeping a constant threshold, and averaged for 111–270 MTs/MT bundles for each Cin8-GFP NL variant concentration. Statistical analysis was performed by one-way ANOVA all pairwise analysis with Tukey correction; DF = 3 and F factor 49.1, and DF = 5 and F factor 70.8 for Cin8 concentrations of 0.17 µg/ml and 0.5 µg/ml, respectively.

## MT sliding assay

The MT sliding assay was performed on piranha-cleaned salinized coverslips as previously described (*Shapira et al., 2017*): Two sets of MTs were polymerized as in the single molecule motility assay, where one of the sets was polymerized without biotinylated tubulin. Then, additional rhodamine-labeled or HILyteFluor488-labeled tubulin (Cytoskeleton TL488M) was added to the polymerizing MTs for 1 hr at 37°C to facilitate bright labeling of the plus end. Cin8 was added in motility buffer, as in the single molecule motility assay, to the immobilized MTs and incubated for 1 min; thereafter MTs in motility buffer were added to the chamber and imaged as described above. One frame of Cin8-GFP was captured, followed by time sequence imaging of 400 s with 10 s intervals of the rhodamine-labeled MT signal. For MT sliding experiments, motor concentration was optimized so that sufficient MT sliding events could be observed, without extensive MT bundling. Concentrations (ng/ml) were as follows (see also *Figure 6B*): wt Cin8 ~3; Cin8-G522N ~110; Cin8$_{NL}$Cut7 ~300; Cin8$_{NL}$Eg5-NG ~45.

## Directionality and velocity analysis of Cin8-induced MT sliding

Kymographs were generated using ImageJ software for Cin8-induced MT sliding with at least one MT being plus-end labeled. Directional Cin8-induced MT sliding was categorized as the mobile MT undergoing a total displacement (starting point to finish point) larger than 0.64 µm and without movements to the opposite direction larger than 0.64 µm and longer than 30% of total time moving. Bi-directional Cin8-induced MT sliding was categorized as exhibiting movements larger than 0.64 µm to both sides, where the time spent moving in the main direction was less than 70% of the total moving time. Plus-end directed velocity was calculated by dividing the distance of continuous displacement on the MT in the plus end direction by the corresponding time. The plus-end directed velocity of 15–24 events was averaged as indicated in *Figure 6C*. Statistical analysis of sliding velocities compared to wt Cin8 was performed by *Dunnett, 1955*; DF = 81, F factors 2.39 ($\alpha$=0.05) and 3.00 ($\alpha$=0.01).

## Doubling time of *S. cerevisiae* cells

The doubling time of *S. cerevisiae* cells was determined as follows: *cin8Δ* or *cin8Δkip1Δcyh$^r$* cells were transformed with a centromeric plasmid expressing one of the Cin8 NL variants and grown overnight. In *cin8Δkip1Δcyh$^r$* cells, the original plasmid encoding a wt Cin8 and a dominant cycloheximide sensitive gene (CYH) (pMA1208) was shuffled out by addition of 7.5 µg/ml cycloheximide to the growth medium. The cells were diluted to OD$_{600}$ ~0.2 2 hr prior to the start of the experiment. The OD$_{600}$ of cells then was measured over time for a culture growing at 26°C, and a plot of $ln\left(\frac{OD_t}{OD_0}\right)$ as a function of time was generated, where OD$_t$ is the OD$_{600}$ at a given time, and OD$_0$ is the OD$_{600}$ at the initial time point. Finally, the doubling time was calculated using the slope of the plot

generated according to the equation: $doublingtime = \frac{ln2}{slope}$. For each variant, the experiment was repeated three to four times. Statistical analysis was performed by *Dunnett, 1955* test compared to wt Cin8; DF = 10, and F factors 2.67 (α=0.05) and 3.77 (α=0.01), and DF = 9, and F factors 2.76 (α=0.05) and 3.74 (α=0.01) for *cin8Δ* and *cin8Δkip1Δ*, respectively.

## Statistical analysis

Data was first examined for normality by the Kolmogorov-Smirnov test (*Massey, 1951*). If normality was rejected, data was subjected to an appropriate Box-Cox transformation (*Sakia, 1992*) to yield normal distributions. Next, significant differences were assessed using ANOVA, followed by either all pairwise comparisons with Tukey correction *Tukey, 1949*, or comparisons to wt Cin8 as a control, according to Dunnett's method *Dunnett, 1955*.

# Acknowledgements

We thank Prof. Levi Gheber, Department of Biotechnology Engineering, BGU, for assistance with image analysis and statistical analysis of the data. We thank Dr. Mary Popov, Dr. Nurit Siegler and Ms. Tatiana Zvagelsky of the LG group for critical reading of this manuscript. This research was supported in part by the Israel Science Foundation grant (ISF-386/18) and the Israel Binational Science Foundation grant (BSF-2015851), awarded to LG.

# Additional information

## Competing interests

Itamar Kass: is affiliated with InterX LTD. The author has no other competing interests to declare. The other authors declare that no competing interests exist.

## Funding

| Funder | Grant reference number | Author |
|---|---|---|
| Israel Science Foundation | ISF-386/18 | Larisa Gheber |
| United States - Israel Binational Science Foundation | BSF-2015851 | Larisa Gheber |

The funders had no role in study design, data collection and interpretation, or the decision to submit the work for publication.

## Author contributions

Alina Goldstein-Levitin, Data curation, Formal analysis, Validation, Investigation, Methodology, Writing - original draft, Writing - review and editing; Himanshu Pandey, Data curation, Formal analysis, Validation, Investigation, Writing - original draft, Writing - review and editing; Kanary Allhuzaeel, Formal analysis, Investigation; Itamar Kass, Investigation, Writing - original draft; Larisa Gheber, Conceptualization, Resources, Formal analysis, Supervision, Funding acquisition, Validation, Methodology, Writing - original draft, Project administration, Writing - review and editing

## Author ORCIDs

Himanshu Pandey (iD) http://orcid.org/0000-0002-0629-7525
Larisa Gheber (iD) https://orcid.org/0000-0003-3759-4001

## Decision letter and Author response

Decision letter https://doi.org/10.7554/eLife.71036.sa1
Author response https://doi.org/10.7554/eLife.71036.sa2

## Additional files

### Supplementary files

• Source data 1. Doubling time (average ± SEM, min) for *S. cerevisiae* cells expressing wt and NL variants of Cin8, presented in *Table 1*.

• Source data 2. Data of three independent experiments for spindle morphology and length distribution of cells expressing NL variants of Cin8. The number of cells and percentage of cells in each category are presented for each Cin8 variant. The average of percentages (± SEM) are presented in *Figure 2A*.

• Source data 3. Area (pixel$^2$) of nuclear localization of 3GFP-tagged wt and NL variants of Cin8. The average area (± SEM) for the number of cells analyzed is presented in *Figure 2E*.

• Source data 4. Data for average (± SEM) number of 3GFP-tagged motors bound per MT length, for wt and NL Cin8 variants, presented in *Figure 4A*.

• Source data 5. Mean displacement (MD), with 1 s time interval, for motility of GFP-tagged Cin8 variants. The plot of MD vs. time interval is presented in *Figure 4D* and the average velocities are obtained from linear fitting the plot to $MD = v \cdot t$. The tab 'Data MD' presents the average MD for each 1 s time interval, up to 8 s, for the indicated Cin8-GFP variants. The tabs 'wt Cin8', 'Cin8-G522N', 'Cin8NLCut7', and 'Cin8NLEg5-N522G' present raw MD data for motility of the corresponding variants.

• Source data 6. Run length (nm) determined for individual GFP-tagged wt and NL variants of Cin8, in the single molecule motility assay. The distribution and average (± SEM) of run lengths are presented in *Figure 4—figure supplement 2B*.

• Source data 7. Intensity of MT bundles (a.u.) as a function of the concentration of Cin8 variants. Average intensities (± SEM) are presented in *Figure 5D*.

• Source data 8. Data for average (± SEM) velocities (µm/min) of MT sliding induced by the different Cin8 variants. MT sliding velocities in plus- and minus-end directions are indicated, data are presented in *Figure 6B*.

• Supplementary file 1. List of plasmids used in this study.

• Supplementary file 2. *Saccharomyces cerevisiae* strains used in this study.

• Transparent reporting form

### Data availability

All data generated or analysed during this study are included in the manuscript and supporting files. All Source data files have been provided.

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
