## [Decision Letter]

**Acceptance summary:**

This paper dissects in great detail the properties of the neck linker of the bidirectional kinesin Cin8. Neck linker is the main mobile element for generating a step by the motor protein kinesin. The described observations on neck linker mutations provide valuable insights into the determinants of bidirectionality of Cin8 and motivate further investigation into the underlying mechanism.

**Decision letter after peer review:**

[Editors’ note: the authors submitted for reconsideration following the decision after peer review. What follows is the decision letter after the first round of review.]

Thank you for submitting your work entitled "Flexibility of the neck-linker during docking is pivotal for function of bi-directional kinesin" for consideration by *eLife*. Your article has been reviewed by 3 peer reviewers, one of whom is a member of our Board of Reviewing Editors, and the evaluation has been overseen by a Senior Editor. The reviewers have opted to remain anonymous.

Our decision has been reached after consultation between the reviewers. Based on these discussions and the individual reviews below, we regret to inform you that your work will not be considered further for publication in *eLife*.

All three reviewers felt that the experiments are appropriate and well-done and that the manuscript is well-written. However, as you will see from the specific comments below, all three reviewers felt the results and interpretations are limited to our understanding of Cin8 motility and do not provide sufficient knowledge to advance our understanding of how some kinesin-5 motors achieve bidirectional motion. All three felt this is a very nice story and should be published with the toning down of some interpretations and we hope the reviewers' comments will be helpful to you in this regard.

*Reviewer #1:*

The manuscript by Goldstein-Levitin et al. examines the role of a kinesin structural element called the neck linker (NL) in the function of an unusual kinesin-5 motor, the S cerevisiae kinesin-5 motor Cin8, which previous work has demonstrated to switch directionality from fast processive minus‐end directed to slow, processive plus‐end directed motility depending on ionic conditions or motor number.

Using both in vitro and in cell assays, the authors demonstrate that replacement of the entire NL by the analogous sequences from either a plus-end motor or bidirectional motor is detrimental to single motor motility and cellular function. In contrast, single point mutations either have no effect or decrease motility properties with minor effects on cellular function. Overall the results provide support for the hypothesis that the NL plays a role in kinesin-5 motility. However, I find the conclusion that "flexibility of the NL during docking is pivotal for function of bi-directional kinesin" unsupported for the reasons outlined below. The authors have carried out the proper experiments and these are well-done but I find the overall conclusions need additional consideration.

1. The authors are equating the presence of a G residue in the N-latch position of Cin8 with flexibility of the NL. It is not clear to me what they mean by flexibility (in the undocked position? in the ionic or backbone interactions in the docked position?). What is the evidence that the flexibility is changed?

2. For Cin8, the authors demonstrate that mutation of G in the N-latch position increases the doubling time but affects cell viability only at high temperatures. The authors conclude that "increased flexibility of the NL during docking of Cin8, compared to Eg5 (Figure 5) provides the necessary adaptation to allow bi-directional motility." (p.16, Discussion). However, the other bi-directional kinesin-5s (Kip1 and Cut7) have an N at the N-latch position and their NLs are thus presumably not flexible although the motors are functional and cells are viable.

3. On p. 14 in the Discussion, the authors state "The profound difference between the activity of Cin8NLEg5 and Cin8NLEg5-NG variants indicates that the glycine in position 522 is critical for the functions of Cin8." The problem is that this conclusion is based on a chimeric motor that does not occur naturally. In fact, if one considers Cin8 with its own NL sequence, both the WT and the G522N mutant Cin8 motors are functional and cells expressing these motors are viable under normal growth conditions. In addition, the other bi-directional kinesin-5s also have an N at this position and are functional.

4. The relationship between G vs N in the N-latch position and motor motility and function is overall unclear. The authors focused on 3 mutant versions in motility assays: Cin8-G522N (has N in N-latch position), Cin8NLCut7 (has N in N-latch position), and Cin8NLEg5-NG (has G in N-latch position). In single-molecule motility assays, all show decreased affinity to the MT and processive minus end-directly motility albeit with decreased velocity as compared to the WT Cin8 motor (has G in N-latch position). In MT bundling assays, all showed a reduced ability to bundle MTs. But their functions are different as are their effects on cell viability.

For example, the NLCut7 and G522N mutants both have an N in the N-latch position. Both show decreased MT affinity, slower minus end-directed motility, and decreased MT bundling activity. Both localize to the SPB in early mitosis. However the NLCut7 mutant generates more monopolar spindles and is not viable at normal yeast growth temperature whereas the G522N mutant generates normal spindle distributions and is viable. It thus appears that having a N in the N-latch position does not correlate with viability.

As a second comparison, the Eg5-NG mutant (G in N-latch position) and the G522N (N in N-latch position) both display decreased MT affinity and slower minus end-directed motility. The Eg5-NG mutant shows severely reduced MT bundling activity and the G522N mutant shows reduced MT bundling activity. Both localize to the SPB in early mitosis and generate normal spindle distributions. And both are viable. It thus appears that the residue in the N-latch position does not correlate with cellular function.

5. In my opinion, it would be most interesting to know whether the NL replacements or mutations alter the bi-directional behavior of Cin8. This is not clear to me from Figure 3. Can the authors provide information about the frequency of each type of motion (fast minus end, diffusive, slow plus end) for each motor at each salt concentration? While all of the mutant motors examined in these assays are slower than the WT motor, their frequency of transitioning to plus end growth is unclear. What salt condition was utilized for the data in 3D?

6. Does mutation of the N>G in the NL of Cut7 in the NLCut7 mutant rescue viability?

*Reviewer #2:*

The authors investigate the role of neck linker docking in the bidirection motility of Cin8. The N-latch idea is well established for kinesin-1 from Lang and Hwang's work. Most kinesin-5 have an N in this position, including two of the three bidirectional kinesin-5, but Cin8 has a glycine in this position. This is notable, because differences in neck linker docking are a good candidate for why fungal kinesin-5 are bidirectional.

The authors investigate chimaeras containing Eg5 or Cut7 neck linker replacement into Cin8. A number of assays are used including viability, in vivo localization, single-molecule velocity, and in vitro microtubule bunding. Pleasingly, the assays all give pretty consistent results. Cin8 doesn't like N replacing G, it can handle the Eg5 neck linker as long as you mutate N to G, and Cut7 NL works better than Eg5.

The paper is well written and the diversity of assays to demonstrate functionality is a strong point of the paper. The question of bidirectionality is important and still puzzling from a mechanistic standpoint.

A weakness is that there is clearly a complex interaction between the neck linker, the cover bundle and the catalytic core of the head, and so it is difficult to interpret what these mutations might be doing to the complex mechanochemistry of these motors. It is notable that the three key residues that interact with N in kinesin-1 are conserved in Cin8, but that means we don't really understand the glycine is doing in wild-type. That mutating it to N screws the motor up is notable, but doesn't shed a lot of light either on NL docking or on bidirectionality.

A smaller but important point is the use of the term "flexibility". Generally glycines increase flexibility of polypeptides. But that isn't actually rigorously shown here. And the effect of the glycine may not have anything to do with flexibility, and rather may all be about hydrogen bonds. In the discussion, the authors bring up a kinesin-1 with longer and hence more flexible neck linker that can be made to step backwards, but that is adding sequence and hence flexibility at the distal part of the neck linker that doesn't dock, so I don't think this comparison is necessarily apt.

The changes in speed are notable, but the mechanism of stepping is too complicated to just attribute the changes to neck linker docking. Because neck linker docking is so fundamental to the chemomechanical cycle, changing these interactions could be changing a lot about the motor.

Overall, this is important work for the field and to add another detail to the question of how fungal kinesin-5 can walk bidirectionally. However, rather than making us think differently about the problem, it is contributing important new details. That limits the impact of the study in my eyes.

*Reviewer #3:*

While kinesin motors have been intensively studied for over thirty years, and while there is consensus on mechanisms underlying "+" end directed movement and processivity, exceptions to the current "rules" exist, most notably concerning the ability of some kinesin 5 family members to move processively in both the "+" and "-" directions. This remarkable feature, which has been known for over a decade, begs a structure-based explanation. In this manuscript, the authors use a mutational approach to examine the structural features needed for bi-directional processive movement in Cin8. Their studies, spanning cell biology to single molecule motillity assays lead them to conclude that bi-directional motility requires "flexibility" in the neck linker, which in Cin8 is provided by the presence of a glycine in a position that in plus end directed kinesins is occupied by an apsaragine. They hypothesize that this single residue difference destabilizes the neck linker from docking along the motor surface toward the "+" end and increases said flexibility.

While the experiments appear to be well done and the text is clearly written, I have several major editorial concerns that I believe need to be addressed in a revised manuscript:

1. NL flexibility: The authors are vague as to what they mean here. One presumes, from their citing of literature that proposes an ordered to disordered NL transition that this flexibility implies increased disorder in the NL which somehow translates to "-" directed motion. However, I note that cryoEM reconstructions of several "+" directed kinesins, including Eg5 and Kif20A, show that the NL is in fact ordered in both pre- and post-hydrolytic states, pointing towards the "+" end in the former and the "-" end in the latter. In view of this, it seems much more likely that there is rather a dynamic equilibrium between two orientations in the NL (minus and plus directed) and that while the NL asparagine drives this equilibrium toward a structured (non flexible) NL in a plus orientation, its loss drives this equilibrium toward a similarly structured, non flexible NL pointing in the minus direction. This is not a minor point, since the term flexibility that the authors use here could mean NL conformational flexibility (ordered versus disordered) or directional flexibility (minus versus plus directed). I believe that the authors need to be considerably clearer about what they mean by "flexibility" and how they envision the loss of the stabilizing hydrogen bonds leading mechanistically to "-" directed motion.

2. Importance of NL glycine: Given the importance the authors place in Cin8 glycine 522 in determining directional flexibility, it is striking that two other bi-directional kinesin 5 family members (Kip1 and Cut7) have an arginine at this position. The authors cite a a cryoEM model (which incidentally while included in the references as #29 is not cited in the text of the discussion) indicating that the NL of cut7 is not stabilized by NL-motor hydrogen bonds to the same degree as wild type Cin8. However, I am not sure that a 4.5 Å resolution map can definitively allow for this conclusion, and I believe that the authors need to expand their discussion of this point, including the potential limitations of their interpretation in this light. Finally, the authors note that mutating the N latch in kinesin 1 increases motor velocity. This is a striking finding that appears to be consistent with their own studies of Cin8. Certainly one possibility is that hydrogen bonding between the NL and the motor core slows NL undocking, which becomes rate limiting in the single molecule assay. Regardless, some mechanism needs to be proposed to explain this besides the relatively uninformative statement that "stabilization of the NL docking plays similar role in controlling the velocity of plus- and minus-end directed N terminal kinesin motors.

---

## [Author Response]

[Editors’ note: The authors appealed the original decision. What follows is the authors’ response to the first round of review.]

Reviewer #1:The manuscript by Goldstein-Levitin et al. examines the role of a kinesin structural element called the neck linker (NL) in the function of an unusual kinesin-5 motor, the S cerevisiae kinesin-5 motor Cin8, which previous work has demonstrated to switch directionality from fast processive minus‐end directed to slow, processive plus‐end directed motility depending on ionic conditions or motor number.Using both in vitro and in cell assays, the authors demonstrate that replacement of the entire NL by the analogous sequences from either a plus-end motor or bidirectional motor is detrimental to single motor motility and cellular function. In contrast, single point mutations either have no effect or decrease motility properties with minor effects on cellular function. Overall the results provide support for the hypothesis that the NL plays a role in kinesin-5 motility. However, I find the conclusion that "flexibility of the NL during docking is pivotal for function of bi-directional kinesin" unsupported for the reasons outlined below. The authors have carried out the proper experiments and these are well-done but I find the overall conclusions need additional consideration.1. The authors are equating the presence of a G residue in the N-latch position of Cin8 with flexibility of the NL. It is not clear to me what they mean by flexibility (in the undocked position? in the ionic or backbone interactions in the docked position?). What is the evidence that the flexibility is changed?

We thank the reviewer for this comment, and we agree that the term “flexibility” that we used to describe NL docking and dynamics could be ambiguous. Similar concerns are also expressed by Reviewer # 2 (comments 1 and 2) and by Reviewer # 3 (comment 1). To address this point, in collaboration with Dr. Itamar Kass, we performed homology modeling of the Cin8 motor domain of wt Cin8 and its NL mutants (Figures 3 and S2 and Table 2). The modeling is based on the following four PDB structures of kinesin motors published previously, all in the presence of AMPPNP—conditions under which the NL is docked and points in the plus-end direction of the MT: (a) 3HQD, X-ray structure of the Eg5 motor domain (Parke et al., 2010); (b) 1VFV, X-ray structure of KIF1A kinesin heavy chain isoform 5C (Nitta et al., 2004); (c) 3ZFC, X-ray structure of KIF4A isoform 4A (Chang et al., 2013); and (d) 6S8M, CryoEM structure of *S. pombe* kinesin-5 Cut7 decorating MTs (von Loeffelholz et al., 2019).

We used the generated model of the Cin8 motor domain to study the H-bond network between residues in the NL (^513^KNIKNKPQLGSF^524^) and in the β7 strand of the motor domain. Calculation of the H-bonds is based on the distance between donor and acceptor atoms (smaller than 0.35 nm) and the angle between the acceptor, donor and hydrogen atoms (smaller than 30 degrees).

Our analysis revealed that mutations in the NL affect the population of H-bonds formed between the NL and β7 of the motor domain. Whereas in wt Cin8 three H-bonds were formed in this region, in variants with partial activity (Cin8-G522N and Cin8_NL_Eg5-NG) only two H-bonds were formed, and in the variant that was severely defective in intracellular functions (Cin8_NL_Cut7) only one H-bond was formed in this region. In addition, a conserved H-bond between the N-latch position (G522) and G157 in a loop between α1 and β3 of the motor domain appears to be critical for Cin8 function. Elimination of this bond resulted in a non-functional variant (Cin8_NL_Cut7-NG). Additional H-bonds stabilizing the N-latch docking to the motor domain (in Cin8_NL_Eg5) also resulted in a non-functional variant. In this variant, replacement of the N-latch asparagine by glycine (as is the case in Cin8) eliminated the additional H-bonds of the N-latch position and rescued the functional defects. Based on this analysis, we conclude that exact H-bond stabilization of the docked NL is critical for the function of Cin8.

Most importantly, the new analysis presented in Figure 3 and Table 2 diverts the discussion from focusing on the N-latch glycine to a more general view of H-bond stabilization involving additional amino acids of the NL (comment # 3 of Reviewer 1; comment # 4 of Reviewer # 2 and comment # 1 of Reviewer # 3). This more general view of NL docking in which the exact H-bond stabilization (at least of certain positions) is critical for the function of a bi-directional kinesin motor may be a general trait of bi-directional motors. As is mentioned above, this notion should be examined experimentally for additional bi-directional kinesin-motors.

2. For Cin8, the authors demonstrate that mutation of G in the N-latch position increases the doubling time but affects cell viability only at high temperatures. The authors conclude that "increased flexibility of the NL during docking of Cin8, compared to Eg5 (Figure 5) provides the necessary adaptation to allow bi-directional motility." (p.16, Discussion). However, the other bi-directional kinesin-5s (Kip1 and Cut7) have an N at the N-latch position and their NLs are thus presumably not flexible although the motors are functional and cells are viable.

As we emphasized in response to the previous point, based on the new structural analysis, in the revised version of the manuscript we do not focus on the N-latch glycine found in Cin8, but generally look at stabilizing H-bond formation between the docked NL and the motor domain. In fact, we observed that in the partially functional NL variants that contain an asparagine in the Nlatch position the number of stabilizing H-bonds (between the N-latch position and part of β10) is smaller than that in wt Cin8, which contains glycine in this position. This is true for the Cin8-G522N, Cin8_NL_Cut7 and Cin8_NL_Eg5-NG variants. Thus, it is not the glycine per se, but the arrangement of the H-bonds, that is important for Cin8 functionality. Based on our new structural analysis, we found a correlation between partial functionality of the variants and the smaller number of stabilizing Hbonds between the docked NL and the motor domain. We discuss this point in the Results section (pages 10-13) and in the discussion (pages 22-23).

3. On p. 14 in the Discussion, the authors state "The profound difference between the activity of Cin8NLEg5 and Cin8NLEg5-NG variants indicates that the glycine in position 522 is critical for the functions of Cin8." The problem is that this conclusion is based on a chimeric motor that does not occur naturally. In fact, if one considers Cin8 with its own NL sequence, both the WT and the G522N mutant Cin8 motors are functional and cells expressing these motors are viable under normal growth conditions. In addition, the other bi-directional kinesin-5s also have an N at this position and are functional.

We thank the reviewer for this comment and agree with the reviewer (and the other reviewers) that not all the intracellular phenotypes are completely consistent with the in vitro data and that in the previous version of the manuscript too much emphasis was put on the N-latch glycine. Based on the new analysis performed on the 3D models (Figure 3 and Table 2), additional amino acids from the NL are involved in the stabilization of the docked NL of some Cin8 mutants. Therefore, we discussed these results from a more general view point (rather than emphasizing only the N-latch glycine in Cin8) (pages 10-13).

That having been said, we also observed that over stabilization of the N-latch position, which is generated by replacing part of the NL sequence of Cin8 with sequences from the NL of the plus-end directed kinesin-5 Eg5, results in a non-functional variant. In this variant, a single replacement of the N-latch asparagine with glycine (as found in Cin8) decreases the number of stabilizing H-bonds at the N-latch position and rescues the functional defects. These data suggest that over stabilization of the N-latch position has a marked effect on the functionality of Cin8. We also found that complete elimination of H-bond stabilization of the N-latch position resulted in a non-functional variant. Thus, we conclude that exact stabilization of the N-latch position is critical for the function of Cin8 (page 11, last paragraph).

4. The relationship between G vs N in the N-latch position and motor motility and function is overall unclear. The authors focused on 3 mutant versions in motility assays: Cin8-G522N (has N in N-latch position), Cin8NLCut7 (has N in N-latch position), and Cin8NLEg5-NG (has G in N-latch position). In single-molecule motility assays, all show decreased affinity to the MT and processive minus end-directly motility albeit with decreased velocity as compared to the WT Cin8 motor (has G in N-latch position). In MT bundling assays, all showed a reduced ability to bundle MTs. But their functions are different as are their effects on cell viability.For example, the NLCut7 and G522N mutants both have an N in the N-latch position. Both show decreased MT affinity, slower minus end-directed motility, and decreased MT bundling activity. Both localize to the SPB in early mitosis. However the NLCut7 mutant generates more monopolar spindles and is not viable at normal yeast growth temperature whereas the G522N mutant generates normal spindle distributions and is viable. It thus appears that having a N in the N-latch position does not correlate with viability.As a second comparison, the Eg5-NG mutant (G in N-latch position) and the G522N (N in N-latch position) both display decreased MT affinity and slower minus end-directed motility. The Eg5-NG mutant shows severely reduced MT bundling activity and the G522N mutant shows reduced MT bundling activity. Both localize to the SPB in early mitosis and generate normal spindle distributions. And both are viable. It thus appears that the residue in the N-latch position does not correlate with cellular function.

In view of the new analysis that we performed, we completely agree with the reviewer that the presence of asparagine in the N-latch position per se does not correlate with cell viability. In the revised version of the manuscript, we show that cell viability correlates with decreased H-bond stabilization of the docked NL. For example, both Cin8-G522N and Cin8_NL_Eg5-NG exhibit reduced Hbond stabilization between the docked NL and β7 of the motor domain (Figure 3 and Table 2), and viability of cells expressing these variants as the sole source of kinesin-5 is reduced (Figure 1C). In Cin8_NL_Cut7 the H-bonds between the docked NL and β7 are eliminated completely, with only one Hbond remaining between the N-latch position and D137, and cell viability is reduced dramatically (Figure 1C). Based on this result, we concluded that the stabilization between docked NL and β7 is important, but not critical for Cin8 function (page 10 and page 11, the first paragraph).

In addition, to address the important point regarding the correlation between motor motility and cell viability, we performed an additional analysis of the run length of single molecules of wt Cin8 and its NL variants (Figure S4). We found that only the Cin8_NL_Cut7 variant exhibits a significantly shorter run length compared to wt Cin8. In addition, the Cin8_NL_Cut7 variant requires a high protein concentration to produce efficient antiparallel MT sliding (Figure 6B), indicating less efficient sliding. Thus, although cell viability is a complex phenomenon, we propose that the significantly reduced viability of cells expressing Cin8_NL_Cut7 results, at least in part, from a combination of different motile defects, which correlate with the reduced H-bond stabilization of NL docking (page 16, last paragraph and page 22, lines 17-25).

Finally, we might have mistakenly implied, in the previous version of the manuscript, that N-latch glycine is the only important factor controlling NL dynamics and Cin8 function, but very likely, this is not the case. In the revised version of the manuscript, we have clarified this point and have emphasized the contribution of other amino acids in the NLs of the mutants that affect NL docking dynamics and function in vivo and in vitro (pages 10-13).

5. In my opinion, it would be most interesting to know whether the NL replacements or mutations alter the bi-directional behavior of Cin8. This is not clear to me from Figure 3. Can the authors provide information about the frequency of each type of motion (fast minus end, diffusive, slow plus end) for each motor at each salt concentration? While all of the mutant motors examined in these assays are slower than the WT motor, their frequency of transitioning to plus end growth is unclear. What salt condition was utilized for the data in 3D?

We thank the Reviewer for this comment. To address this important comment, we examined the directionality of antiparallel MT sliding induced by wt Cin8 and the different NL variants. It has previously been shown that one of the conditions under which Cin8 reverses directionality from minus- to plus-end directed motility occurs when crosslinking two antiparallel MTs and mediating their sliding apart by plus-end directed motility on the two MTs (Gerson-Gurwitz et al., 2011; Roostalu et al., 2011; Shapira et al., 2017). In addition, we note that – based on our recently published model (Shapira et al., 2017) – directionality reversal to plus-end directed motility during antiparallel MT sliding is an important motor function essential for mitotic spindle assembly.

In the current study, MT sliding experiments were performed using polarity-marked MTs, as previously described (Shapira et al., 2017). Surprisingly, we observed two different MT sliding modes (Figures 6 and S5): plus-end directed sliding, characterized by MT sliding with its unlabeled minus-end leading (Figure 6), and bi-directional/minus-end directed sliding, characterized by at least one episode of minus-end directed MT sliding, with its labeled plus-end leading (Figure S5). Our data indicates that all the functional variants that can support cell viability as single source for kinesin-5 function can produce plus-end directed antiparallel sliding. The occurrence of such sliding was similar for all the examined variants. These findings support the notion that plus-end directed antiparallel sliding is required for intracellular kinesin-5 function.

For the partially active variants, the motor protein concentration under which antiparallel MT sliding was observed was considerably higher than that for wt Cin8. This result is consistent with the lower ability of the NL variants to crosslink antiparallel MTs (Figure 5). Finally, the velocity of MT sliding was slowest when mediated by the Cin8_NL_Cut7 and Cin8_NL_Eg5-NG variants, which can account, in part, for the reduced functionality of these variants in cells (Figure 1C).

6. Does mutation of the N>G in the NL of Cut7 in the NLCut7 mutant rescue viability?

We thank the Reviewer for this interesting question. To address this point, we generated the Cin8_NL_Cut7-NG variant, which contains the NL of Cut7 but with the N-latch asparagine of Cin8replaced with glycine. Surprisingly, we found that, in contrast to the Cin8_NL_Eg5-NG variant, the Cin8_NL_Cut7-NG variant was completely non-functional in cells (Figure 1C); it accumulated with monopolar spindles (Figure 2A) and exhibited diffusive localization in the nucleus (Figure 2C-E). This variant also exhibited a significantly reduced MT-binding ability (Figure 4A). Examination of the H-bond configuration of this variant revealed that it failed to form a conserved backbone H-bond between the N-latch position and a glycine in a loop between α1 and β3 of the motor domain of Cin8 (G157) (Figure 3 and Table 2). Based on this result, we conclude that this conserved backbone H-bond is critical for Cin8 function (page 9 last paragraph, page 22, lines 26-31).

Reviewer #2:The authors investigate the role of neck linker docking in the bidirection motility of Cin8. The N-latch idea is well established for kinesin-1 from Lang and Hwang's work. Most kinesin-5 have an N in this position, including two of the three bidirectional kinesin-5, but Cin8 has a glycine in this position. This is notable, because differences in neck linker docking are a good candidate for why fungal kinesin-5 are bidirectional.The authors investigate chimaeras containing Eg5 or Cut7 neck linker replacement into Cin8. A number of assays are used including viability, in vivo localization, single-molecule velocity, and in vitro microtubule bunding. Pleasingly, the assays all give pretty consistent results. Cin8 doesn't like N replacing G, it can handle the Eg5 neck linker as long as you mutate N to G, and Cut7 NL works better than Eg5.The paper is well written and the diversity of assays to demonstrate functionality is a strong point of the paper. The question of bidirectionality is important and still puzzling from a mechanistic standpoint.

We thank the Reviewer for these comments.

A weakness is that there is clearly a complex interaction between the neck linker, the cover bundle and the catalytic core of the head, and so it is difficult to interpret what these mutations might be doing to the complex mechanochemistry of these motors. That mutating it to N screws the motor up is notable, but doesn't shed a lot of light either on NL docking or on bidirectionality.A smaller but important point is the use of the term "flexibility". Generally glycines increase flexibility of polypeptides. But that isn't actually rigorously shown here. And the effect of the glycine may not have anything to do with flexibility, and rather may all be about hydrogen bonds. In the discussion, the authors bring up a kinesin-1 with longer and hence more flexible neck linker that can be made to step backwards, but that is adding sequence and hence flexibility at the distal part of the neck linker that doesn't dock, so I don't think this comparison is necessarily apt.

We fully agree with the Reviewer that the interaction of the NL with the catalytic motor domain is complex and that focusing on the N-latch glycine/asparagine of Cin8 was an oversimplification. We also agree with comment #2 in that the term “flexibility” was poorly defined and characterized and that stabilization by H-bonds should have been considered in the previous version of the manuscript. To address this issue (and similar comments raised by Reviewers #1 and #3), we generated homology models of the Cin8 motor domain of wt Cin8 and its NL mutants (Figure 3, S2 and Table 21). Our modeling is based on four PDB structures published previously for kinesin motors; please see a detailed description in our response to comment #1 of Reviewer #1. We used the generated model of the Cin8 motor domain to calculate the H-bonds between the NL (^513^KNIKNKPQLGSF^524^) and the motor domain (Figure 3, S2 and Table 2 of the revised manuscript).

Our analysis revealed that in wt Cin8 one conserved backbone H-bond is formed between the Nlatch G522 and G157. Based on the study of Hwang at al. (2008), two backbone H-bonds are formed between the N-latch asparagine and the motor domain, one with a glycine homologous to G157 of Cin8 and one with β7 of the motor domain. This suggests a reduced H-bond stabilization of the Nlatch position of Cin8, compared to kinesin-1. Importantly, our analysis revealed that the number of H-bonds formed between the NL and the motor domain is different from that between wt Cin8 and NL-mutants. In all the partially functional variants, the number of H-bonds formed between the Nlatch position and β10 of the NL and the motor domain is smaller than that in wt Cin8. In the Cin8_NL_Cut7 variant, which exhibits severely defective activity (Figure 1,2), no H-bonds are formed in this region, indicating that these H-bonds are important but not critical for Cin8 function. Finally, our analysis showed that complete elimination of H-bonds at the N-latch position and overstabilization of this position in the docked NL led to the production of the non-functional variants, Cin8_NL_Cut7-NG and Cin8_NL_Eg5, respectively. In the Cin8_NL_Eg5 variant, replacement of the N-latch asparagine with glycine, as in Cin8, eliminated additional H-bonds at this position and rescued the majority of functional defects. Thus, we conclude that exact stabilization of the N-latch position is critical for the function of the bi-directional Cin8.

Finally, regarding the flexibility of NL docking and bi-directional motility of Cin8: Based on our modeling and H-bond analysis, we observed reduced H-bond stabilization of Cin8 compared to kinesin-1 (Hwang et al., 2008). In addition, overstabilization of this position by additional H-bonds resulted in a non-functional variant. Therefore, a certain degree of flexibility is important in this position. Although this “flexibility” may be different for different bi-directional kinesins, in general terms, the reduced stabilization of NL docking that we observed in Cin8 can, in fact, constitute the molecular mechanism required for the bi-directional stepping (see the last section of the Discussion).

It is notable that the three key residues that interact with N in kinesin-1 are conserved in Cin8, but that means we don't really understand the glycine is doing in wild-type.

Although in amino acid sequence alignments, presented in the previous version of the manuscript, it appears that glycine occupies the N-latch position only in Cin8 (Figure 2), in fact, there are other kinesin-5 motors that contain glycine in this position (Figure S1). Therefore, the mechanism of NL docking and dynamics applicable to Cin8 is not unique to Cin8 and probably represents a more general mechanism. In addition, as discussed in response to the previous point, the N-latch glycine of Cin8 does form one of the backbone H-bonds reported for kinesin-1 (Hwang et al., 2008), which indicates that, to some degree, the molecular mechanism involving the N-latch asparagine in kinesin-1 is recapitulated in the N-latch glycine of Cin8.

The changes in speed are notable, but the mechanism of stepping is too complicated to just attribute the changes to neck linker docking. Because neck linker docking is so fundamental to the chemomechanical cycle, changing these interactions could be changing a lot about the motor.

We agree with the Reviewer that the H-bond stabilization mechanism discussed here (Figure 3 and Table 2) – or the “flexibility” discussed in the previous version of the manuscript – do not exclude other structural-functional mechanisms that contribute to NL docking and dynamics. For example, the formation of a β-sheet cover strand between the non-motor N-terminal sequences and the β9 of the NL is probably important in the stepping of Cin8, similarly to other kinesin motors (Goulet et al., 2012; Goulet et al., 2014; Hwang et al., 2008; Khalil et al., 2008; von Loeffelholz et al., 2019) (please see page 10, lines 12-28). Based on our new structural analysis (Figure 3 and Table 2), there are differences in the H-bond arrangement between wt Cin8 and the NL variants. In all the partially functional variants (Cin8-G522N, Cin8_NL_Eg5-NG and Cin8NLCut7), the number of H-bonds formed in this region is smaller, indicating a correlation between H-bond stabilization and Cin8 functionality.

In addition, the new results presented in the revised manuscript indicate that the exact stabilization of the N-latch position is critical for Cin8 function, with elimination and over-stabilization in this position resulting in non-functional Cin8 variants (Cin8_NL_Eg5 and Cin8_NL_Cut7-NG). In the nonfunctional CIn8_NL_Eg5, a single replacement of the N-latch asparagine with glycine, as is present in Cin8, reduces the number of stabilizing H-bonds of the N-latch position and rescues the majority of functional defects in vivo and in vitro. Thus, although other factors undoubtedly affect motor stepping, the strong correlation between H-bond formation and functionality of Cin8 indicates that this is an important factor controlling NL dynamics, motor activity and intracellular functions. As demonstrated here for Cin8, these principles may apply to other bi-directional kinesin motors.

Overall, this is important work for the field and to add another detail to the question of how fungal kinesin-5 can walk bidirectionally. However, rather than making us think differently about the problem, it is contributing important new details. That limits the impact of the study in my eyes.

The new analysis that we performed (Figure 3 and Table 2) demonstrates how H-bond formation regulates the functionality of a bi-directional kinesin. The current paper is thus the first report of such a structural analysis, coupled with experimental data, for a bi-directional kinesin. Our work establishes two new rules for NL-docking of Cin8: (1) the H-bonds formed between the NL and β7 of the motor domain are important but not critical for the function of Cin8; and (2) the exact Hbond stabilization of the N-latch position is critical for the function of Cin8. These factors, especially the second one, are likely to be important for bi-directional stepping. Cin8 serves as an example of a bi-directional kinesin, and more work on additional bi-directional motors is required to establish the generality of these principles (please see the last section of the Discussion). In this respect, the current study is not just another interesting detail, but adds to our understanding of what is required for bi-directional stepping.

Reviewer #3:While kinesin motors have been intensively studied for over thirty years, and while there is consensus on mechanisms underlying "+" end directed movement and processivity, exceptions to the current "rules" exist, most notably concerning the ability of some kinesin 5 family members to move processively in both the "+" and "-" directions. This remarkable feature, which has been known for over a decade, begs a structure-based explanation. In this manuscript, the authors use a mutational approach to examine the structural features needed for bi-directional processive movement in Cin8. Their studies, spanning cell biology to single molecule motillity assays lead them to conclude that bi-directional motility requires "flexibility" in the neck linker, which in Cin8 is provided by the presence of a glycine in a position that in plus end directed kinesins is occupied by an apsaragine. They hypothesize that this single residue difference destabilizes the neck linker from docking along the motor surface toward the "+" end and increases said flexibility.While the experiments appear to be well done and the text is clearly written, I have several major editorial concerns that I believe need to be addressed in a revised manuscript:

We thank the Reviewer for this comment and for stating that we should submit a revised version of this manuscript.

1. NL flexibility: The authors are vague as to what they mean here. One presumes, from their citing of literature that proposes an ordered to disordered NL transition that this flexibility implies increased disorder in the NL which somehow translates to "-" directed motion. However, I note that cryoEM reconstructions of several "+" directed kinesins, including Eg5 and Kif20A, show that the NL is in fact ordered in both pre- and post-hydrolytic states, pointing towards the "+" end in the former and the "-" end in the latter. In view of this, it seems much more likely that there is rather a dynamic equilibrium between two orientations in the NL (minus and plus directed) and that while the NL asparagine drives this equilibrium toward a structured (non flexible) NL in a plus orientation, its loss drives this equilibrium toward a similarly structured, non flexible NL pointing in the minus direction. This is not a minor point, since the term flexibility that the authors use here could mean NL conformational flexibility (ordered versus disordered) or directional flexibility (minus versus plus directed). I believe that the authors need to be considerably clearer about what they mean by "flexibility" and how they envision the loss of the stabilizing hydrogen bonds leading mechanistically to "-" directed motion.

We fully agree with the Reviewer that the term “flexibility” was poorly defined and characterized by us. To address this comment (and similar comments of the other Reviewers), we have now performed additional structural analysis, based on homology modeling of the Cin8 motor domain, to examine H-bond formation between the NL (^513^KNIKNKPQLGSF^524^) and the motor domain. This analysis reveals that, as the reviewer suggested, addition of H-bonds stabilizing the docked N-latch position (such as in the Cin8_NL_Eg5 variant) correlates with the abolished functionality (Figure 1, 3 and Table 1 and 2). (Please also see our response to comment # 1 of Reviewer # 1 and comments # 1 and 2 of Reviewer # 2). The new analysis also reveals that there is a correlation between decrease in stabilizing H-bonds between β10 of the NL and β7 of the motor domain and decreased functionality (please see pages 10-13). These results shift the emphasis to a broader view of H-bond formation. We did find that in the non-functional Cin8_NL_Eg5 variant, which forms additional H-bonds to stabilize the N-latch position, replacing the N-latch asparagine with glycine, as is present in Cin8, reduces the number of stabilizing H-bonds and rescues the majority of functional defects. These results, in fact, support the view outlined here by the Reviewer that there might be a dynamic equilibrium between two conformations of the NL and that additional H-bonds in the non-functional variant shift this equilibrium towards one of the conformations, which interferes with the function of Cin8. In the revised manuscript, we discuss this mechanism suggested by the Reviewer (Page 24, lines 12-20).

2. Importance of NL glycine: Given the importance the authors place in Cin8 glycine 522 in determining directional flexibility, it is striking that two other bi-directional kinesin 5 family members (Kip1 and Cut7) have an arginine at this position. The authors cite a a cryoEM model (which incidentally while included in the references as #29 is not cited in the text of the discussion) indicating that the NL of cut7 is not stabilized by NL-motor hydrogen bonds to the same degree as wild type Cin8. However, I am not sure that a 4.5 Å resolution map can definitively allow for this conclusion, and I believe that the authors need to expand their discussion of this point, including the potential limitations of their interpretation in this light.

We thank the Reviewer for this question. To address this point and to obviate reliance on one structure of 4.5Å resolution, we generated 3D models of the Cin8 motor domain, including the NL (Figure 3 of the revised manuscript). The models were based on four PDB structures published previously for kinesin motors, all in the presence of AMP-PNP, namely, conditions under which the NL is docked and points in the plus-end direction of the MT: (a) 3HQD, X-ray structure of Eg5 motor domain (Parke et al., 2010); (b) 1VFV, X-ray structure of KIF1A kinesin heavy chain isoform 5C (Nitta et al., 2004); (c) 3ZFC, X-ray structure of KIF4A isoform 4A (Chang et al., 2013); and (d) 6S8M, cryo-EM structure of *S. pombe* kinesin-5 Cut7 decorating MTs (von Loeffelholz et al., 2019). Such modeling increases the reliability of our conclusions. (Please also see our response to point # 1of reviewer # 1).

It should be noted that there are other kinesin-5 motors that contain glycine at the N-latch position (Figure S1), which suggests that the NL functionality involving this glycine may be more general and not apply only to one protein. However, in the revised version of the paper, we focus on the stabilization of the NL via H-bond formation and not on a specific sequence. In fact, we show that based on our models, some variants that contain N-latch asparagine, instead of the glycine that is present in Cin8, form a smaller number of H-bonds between NL and β7, compared to wt Cin8 (such as Cin8-G522N, Cin8_NL_Cut7 and Cin8_NL_Eg5-NG; Figure 3 and Table 2 of the revised version). Although we believe that the high “sensitivity” to precise H-bond stabilization of certain positions in the NL that we have demonstrated here for Cin8 is a common trait of bi-directional kinesin motors, it is likely that the precise details and the amino acids that participate in such stabilization are different for the different motor proteins. Additional experiments are needed to examine this mechanism for other bi-directional kinesin motors.

Finally, the authors note that mutating the N latch in kinesin 1 increases motor velocity. This is a striking finding that appears to be consistent with their own studies of Cin8. Certainly one possibility is that hydrogen bonding between the NL and the motor core slows NL undocking, which becomes rate limiting in the single molecule assay. Regardless, some mechanism needs to be proposed to explain this besides the relatively uninformative statement that "stabilization of the NL docking plays similar role in controlling the velocity of plus- and minus-end directed N terminal kinesin motors.

We fully agree with the Reviewer on this point. We have performed structural analysis in order to be more precise in our statements and claims regarding stabilization (vs. flexibility) of certain configurations of the NL. We also referred to the mechanism suggested by this Reviewer in the first comment to explain the reduction in minus-end directed velocity when the docked NL is stabilized by H-bonds (Page 24, lines 12-20).

Barber-Zucker, S., R. Uebe, G. Davidov, Y. Navon, D. Sherf, J.H. Chill, I. Kass, R. Bitton, D. Schüler, and R. Zarivach. 2016. Disease-Homologous Mutation in the Cation Diffusion Facilitator Protein MamM Causes Single-Domain Structural Loss and Signifies Its Importance. *Sci Rep*. 6.

Chang, Q., R. Nitta, S. Inoue, and N. Hirokawa. 2013. Structural basis for the ATP-induced isomerization of kinesin. *Journal of molecular biology*. 425:1869-1880.

Fodor, J., B.T. Riley, I. Kass, A.M. Buckle, and N.A. Borg. 2019. The Role of Conformational Dynamics in Abacavir-Induced Hypersensitivity Syndrome. *Sci Rep*. 9:019-47001.

Gerson-Gurwitz, A., C. Thiede, N. Movshovich, V. Fridman, M. Podolskaya, T. Danieli, S. Lakamper, D.R. Klopfenstein, C.F. Schmidt, and L. Gheber. 2011. Directionality of individual kinesin-5 Cin8 motors is modulated by loop 8, ionic strength and microtubule geometry. *The EMBO journal*. 30:4942-4954.

Goulet, A., W.M. Behnke-Parks, C.V. Sindelar, J. Major, S.S. Rosenfeld, and C.A. Moores. 2012. The structural basis of force generation by the mitotic motor kinesin-5. *J Biol Chem*. 287:4465444666.

Goulet, A., J. Major, Y. Jun, S.P. Gross, S.S. Rosenfeld, and C.A. Moores. 2014. Comprehensive structural model of the mechanochemical cycle of a mitotic motor highlights molecular adaptations in the kinesin family. *Proc Natl Acad Sci U S A*. 111:1837-1842.

Hwang, W., M.J. Lang, and M. Karplus. 2008. Force generation in kinesin hinges on cover-neck bundle formation. *Structure*. 16:62-71.

Kass, I., D.E. Hoke, M.G.S. Costa, C.F. Reboul, B.T. Porebski, N.P. Cowieson, H. Leh, E. Pennacchietti, J. McCoey, O. Kleifeld, C. Borri Voltattorni, D. Langley, B. Roome, I.R. Mackay, D. Christ, D. Perahia, M. Buckle, A. Paiardini, D. De Biase, and A.M. Buckle. 2014. Cofactor-dependent conformational heterogeneity of GAD65 and its role in autoimmunity and neurotransmitter homeostasis. *Proc Natl Acad Sci U S A*. 111:E2524-2529.

Khalil, A.S., D.C. Appleyard, A.K. Labno, A. Georges, M. Karplus, A.M. Belcher, W. Hwang, and M.J. Lang. 2008. Kinesin's cover-neck bundle folds forward to generate force. *Proceedings of the National Academy of Sciences*. 105:19247-19252.

Nitta, R., M. Kikkawa, Y. Okada, and N. Hirokawa. 2004. KIF1A alternately uses two loops to bind microtubules. *Science*. 305:678-683.

Parke, C.L., E.J. Wojcik, S. Kim, and D.K. Worthylake. 2010. ATP Hydrolysis in Eg5 Kinesin Involves a Catalytic Two-water Mechanism. *J Biol Chem*. 285:5859-5867.

Roostalu, J., C. Hentrich, P. Bieling, I.A. Telley, E. Schiebel, and T. Surrey. 2011. Directional switching of the Kinesin cin8 through motor coupling. *Science*. 332:94-99.

Shapira, O., A. Goldstein, J. Al-Bassam, and L. Gheber. 2017. A potential physiological role for bidirectional motility and motor clustering of mitotic kinesin-5 Cin8 in yeast mitosis. *J Cell Sci*. 130:725-734.

Singh, S.K., H. Pandey, J. Al-Bassam, and L. Gheber. 2018. Bidirectional motility of kinesin-5 motor proteins: structural determinants, cumulative functions and physiological roles. *Cell Mol Life Sci*:1757-1771.

von Loeffelholz, O., A. Pena, D.R. Drummond, R. Cross, and C.A. Moores. 2019. Cryo-EM Structure (4.5-A) of Yeast Kinesin-5-Microtubule Complex Reveals a Distinct Binding Footprint and Mechanism of Drug Resistance. *Journal of molecular biology*. 431:864-872.